# Optimization of snow-related parameters in the Noah land surface model (v3.4.1) using a micro-genetic algorithm (v1.7a)

**Sujeong Lim**[1,2]**, Hyeon-Ju Gim**[3]**, Ebony Lee**[1,2,4]**, Seungyeon Lee**[1,2,4]**, Won Young Lee**[1,2]**, Yong Hee Lee**[5]**, Claudio Cassardo**[6]**, and Seon Ki Park**[1,2,4]

[1]Center for Climate/Environment Change Prediction Research, Ewha Womans University, Seoul, 03760, Republic of Korea
[2]Severe Storm Research Center, Ewha Womans University, Seoul, 03760, Republic of Korea
[3]Korea Institute of Atmospheric Prediction System (KIAPS), Seoul, 07071, Republic of Korea
[4]Department of Climate and Energy System Engineering, Ewha Womans University, Seoul, 03760, Republic of Korea
[5]High Impact Weather Research Department, National Institute of Meteorological Sciences,
Gangneung, 25457, Republic of Korea
[6]Department of Physics and NatRisk Centre, University of Turin, Turin, 10125, Italy

**Correspondence:** Seon Ki Park (spark@ewha.ac.kr)

**Abstract.** Snowfall prediction is important in winter and early spring because snowy conditions generate enormous economic damages. However, there is a lack of previous studies dealing with snow prediction, especially using land surface models (LSMs). Numerical weather prediction models directly interpret the snowfall events, whereas LSMs evaluate the snow cover, snow albedo, and snow depth through interaction with atmospheric conditions. Most LSMs include parameters based on empirical relations, resulting in uncertainties in model solutions. When the initially developed empirical parameters are local or inadequate, we need to optimize the parameter sets for a certain region. In this study, we seek the optimal parameter values in the snow-related processes – snow cover, snow albedo, and snow depth – of the Noah LSM, for South Korea, using the micro-genetic algorithm and the in situ surface observations and remotely sensed satellite data. Snow data from observation stations representing five land cover types – deciduous broadleaf forest, mixed forest, woody savanna, cropland, and urban and built-up lands – are used to optimize five snow-related parameters that calculate the fractional snow cover, maximum snow albedo of fresh snow, and fresh snow density associated with the snow depth. Another parameter, reflecting the dependence of fractional snow cover on the land cover types, is also optimized. Optimization of these six snow-related parameters has led to improvement in the root mean squared errors by 17.0 %, 6.2 %, and 3.3 % in snow depth, snow albedo, and fractional snow cover, respectively. In terms of the mean bias, the underestimation problems of snow depth and overestimation problems of snow albedo have been alleviated through optimization of parameters calculating the fresh snow by about 44.2 % and 31.0 %, respectively.

## 1 Introduction

Land surface models (LSMs) act as the lower boundary conditions for regional numerical weather prediction (NWP) and climate models, to which they provide the surface fluxes (Ek et al., 2003). However, LSMs include inevitable uncertainties due to insufficient knowledge of surface layer processes and characteristics; for instance, unreasonable representation of the spatiotemporal surface heterogeneity and the inaccuracy of the parameters based on empirical relations contribute to the uncertainties in LSMs. In particular, uncertainties in the snow-related processes of LSMs are appreciable and exert significant impacts on the performance of regional climate models to which the LSMs are coupled (e.g., Zhao and Li, 2015; Suzuki and Zupanski, 2018; Günther et al., 2019; Kim and Park, 2019; Xu et al., 2019; Jiang et al., 2020).

Intense snowfall events often occur on the Korean Peninsula during winter and early spring. In South Korea (SK), heavy snowfalls are the third-most serious source of natural disasters, following typhoons and heavy rainfalls (Kim et al., 2018), with severe economic consequences. Most of the previous studies focused on classification of snowfall (Cheong et al., 2006), investigation of synoptic characteristics (Jung et al., 2012), and comparisons of different LSM options in the coupled atmosphere–land surface prediction system (Wang and Sun, 2018; Kim and Park, 2019). Being coupled to the atmospheric models, the LSMs play an important role in predicting the snowfall in NWP because they calculate the fractional snow cover, snow albedo, and snow depth through interactions with the atmosphere. For example, the choice of land surface scheme is crucial for simulating the spatial distributions of snowfall in the land-surface-coupled NWP models (e.g., Wang and Sun, 2018; Kim and Park, 2019). In other words, the numerical snowfall forecast is strongly affected by the performance of the coupled LSM; thus, improvement in the snow process parameterizations of the offline LSMs can bring about better performance in NWP models.

Uncertainties in parameterized physical processes have been observed and quantified in various numerical models (e.g., Mallet and Sportisse, 2006; Gubler et al., 2012; Shutts and Pallarès, 2014; Folberth et al., 2019; Li et al., 2020; Olafsson and Bao, 2020; Pathak et al., 2020; Souza et al., 2020). Such uncertainties can be reduced by estimating optimal parameter values in the subgrid-scale parameterization schemes (e.g., Annan and Hargreaves, 2004; Lee et al., 2006; Neelin et al., 2010; Yu et al., 2013; Zhang et al., 2015; Kotsuki et al., 2018; Li et al., 2018; Chinta and Balaji, 2020). Because empirical parameters are commonly derived from the observations or theoretical calculations, their estimated values are strongly dependent on the local characteristics of the region and period where the observations are made. Thus, *parameter estimation* that fits the model outputs to the observations is essentially required to obtain an adequate parameter (Duan et al., 2017). It may be done using a *trial-and-error* manual approach, but the *optimization algorithm* helps to replace enormous experiments by automatically minimizing the difference between model and observations (Duan et al., 2006). For example, a global optimization tool, called the micro-genetic algorithm (micro-GA), has been effectively used for estimating the optimal parameter values in the NWP model (e.g., Yu et al., 2013).

Most snow processes in the LSMs are parameterized based on the observations in specific local regions, and hence they may not represent adequately the situation in SK and be the source of uncertainties for numerical snow prediction over SK. We aim at obtaining the optimal parameter values of the snow-related processes – snow cover, snow albedo, and snow depth – in a LSM using the micro-GA, which causes a better LSM performance over SK. This study represents the first attempt to develop a coupled system of the micro-GA and Noah LSM for parameter estimation of the snow pro-

cesses. Section 2 describes the methodology, including the snow processes of the LSM and the micro-GA optimization tool. Section 3 explains the experiment design. Results and the conclusions and outlook are provided in Sects. 4 and 5, respectively.

## 2 Methodology

### 2.1 Snow-related processes in the Noah land surface model

In this study, we employ the Noah LSM (Chen et al., 1996; Koren et al., 1999; Ek et al., 2003) to simulate the single-site land surface processes (Mitchell, 2005), including the surface energy and water flux, and to verify energy and water budgets in the near-surface atmospheric layer by simulating the soil moisture, soil temperature, and snowpack. The Noah LSM is a stand-alone and one-dimensional column model, developed through multi-institutional cooperation. In the soil, to simulate soil moisture and soil temperature, we selected four layers with thicknesses of 10, 30, 60, and 100 cm, respectively, from top to bottom, for a total depth of 2 m. The model also evaluates various other variables, including skin temperature, snow depth, snow water equivalent, snow density, and canopy water content (Mitchell, 2005). The energy and water fluxes are calculated through the surface energy and water balance equations, respectively. Due to its adequate complexity and computational efficiency (Mitchell et al., 2004), the Noah LSM has been coupled to the operational NWP model of the Korea Meteorological Administration (KMA), named the Korean Integrated Model (KIM; Hong et al., 2018) – see Koo et al. (2017) for details of the coupled KIM–Noah LSM system.

The current Noah LSM (version 3.4.1) uses a single-layer representation of the snow processes considering a bulk snow–soil canopy layer (Sultana et al., 2014). If air temperature is less than $0\,°C$, the resulting precipitation is considered snow. The fractional snow cover is determined as a function of snow water equivalent (SWE) using a generalized snow depletion curve. Snow albedo is calculated based on the fractional snow cover and the maximum snow albedo (Ek et al., 2003). Snow depth is represented by SWE and the bulk snow density (Jonas et al., 2009). The equations in the Noah LSM describe the heat exchanges at the snow–atmosphere and snow–soil interfaces as well as snow accumulation, sublimation, and melting (Suzuki and Zupanski, 2018). The above-mentioned snow processes contain certain estimated coefficients or constants, known as *parameters*, which employ typical, empirical, or a priori values. The parameters are provided as lookup tables based on their samples in the field or lab. Traditionally, they are tuned by trial and error to calibrate the model against historical observations in a specific location; however, a systematic and objective procedure is essentially required for a large number of stations (Duan et al.,

2006; Rosolem et al., 2013). We explain below the details of the snow-related parameters to be optimized for various stations in SK.

### 2.1.1 Fractional snow cover (FSC)

The FSC ($\sigma_\mathrm{s}$) is important for the accumulation and ablation processes (Livneh et al., 2010). As a function of SWE ($W_\mathrm{s}$; in meters) extracted by the atmospheric input values (Livneh et al., 2010), $\sigma_\mathrm{s}$ varies nonlinearly as in Eq. (1), following the empirical snow depletion curves of Anderson (1973):

$$\sigma_\mathrm{s} = 1 - e^{-P_\mathrm{s}W} + We^{-P_\mathrm{s}}. \tag{1}$$

Here, $P_\mathrm{s}$ is the distribution shape parameter, and $W = W_\mathrm{s}/W_\mathrm{max}$, where $W_\mathrm{max}$ is the threshold of $W_\mathrm{s}$ above which $\sigma_\mathrm{s}$ is 100 %. Note that, from Eq. (1), $\sigma_\mathrm{s}$ is a function of $P_\mathrm{s}$ and $W_\mathrm{max}$ – these two parameters are to be optimized.

Figure 1 represents the responses of the snow variables to the variations in the snow-related parameters for given ranges. It is noteworthy that $P_\mathrm{s}$ has a positive correlation with snow cover (Fig. 1a). For example, $\sigma_\mathrm{s}$ increases as $P_\mathrm{s}$ increases, resulting in relatively slow snowmelt. In Eq. (1), the value of $P_\mathrm{s}$ usually ranges between 2 and 4 (e.g., Anderson, 1973; Koren et al., 1999), and its default value in the Noah LSM is 2.6. We seek the optimal value of $P_\mathrm{s}$, which lies between 2 and 4 and is suited to SK.

The SWE threshold, $W_\mathrm{max}$, has a negative correlation with snow cover, as shown in Eq. (1), and it is more sensitive compared to $P_\mathrm{s}$ within a given parameter's range (Fig. 1b). In the Noah LSM, the values of $W_\mathrm{max}$ are prespecified in a table (VEGPARM.TBL), varying with the land cover types (LCTs). $W_\mathrm{max}$ has the largest value over forest, reflecting the irregular geometry of forest cover (Livneh et al., 2010). Previous studies suggest the uncertainty range in the values of $W_\mathrm{max}$; for instance, Livneh et al. (2010) used 0.04 m for forest and 0.02 m for non-forest, respectively, whereas Wang and Zeng (2010) used 0.2 m for tall vegetation and 0.01 m for short vegetation. The default values in the Noah LSM are 0.08 m for forest and 0.04 m for non-forest. We estimate the optimal $W_\mathrm{max}$ values, suited to SK, in the range between 0.01 and 2 m.

### 2.1.2 Snow albedo (SA)

SA is defined as the fraction of incident radiation reflected by the snowpack and is crucial for evaluating surface-energy balance, particularly during snowmelt (Warren and Wiscombe, 1980; Warren, 1982); however, accurate representation of SA is difficult due to numerous complexities (Livneh et al., 2010).

Surface albedo generally increases over snow, but it may react differently over a shallow snowpack: when accumulation starts by snowfall or diminution occurs by snowmelt, patchy areas can be generated and corresponding model grid boxes may not be covered by snow (Ek et al., 2003). The Noah LSM reflects this patchiness effect by calculating surface albedo ($\alpha$) as a composite of snow-covered surface albedo ($\alpha_\mathrm{s}$) and snow-free surface albedo ($\alpha_0$) as

$$\alpha = \alpha_0 + \sigma_\mathrm{s}(\alpha_\mathrm{s} - \alpha_0). \tag{2}$$

Note that SA is generally highest over the fresh snow and decays thereafter, and the decay rate depends on the seasonal snow phase – faster during the ablation phase and slower during the accumulation phase. By reflecting this fact, $\alpha_\mathrm{s}$ is evaluated as a function of the fresh SA ($\alpha_\mathrm{max}$), the number of days after the last snowfall ($t$), and the albedo-decay rates ($A$ and $B$) as

$$\alpha_\mathrm{s} = \alpha_\mathrm{max}A^{t^B}, \tag{3}$$

where the default values of empirical parameters $A$ and $B$ are 0.94 and 0.58, respectively, during the accumulation phase, and 0.82 and 0.46, respectively, during the ablation. However, the current Noah LSM activates only the accumulation phase in Eq. (3), and both $A$ and $B$ are excluded from our optimization.

Spatial variation in SA is taken into consideration in $\alpha_\mathrm{max}$ by incorporating the satellite-based maximum SA ($\alpha_\mathrm{max,sat}$) from Robinson and Kukla (1985) and by imposing adjustment to a maximum SA ($\alpha_\mathrm{max,CofE}$) from USACE (1956) (see also Livneh et al., 2010) as

$$\alpha_\mathrm{max} = \alpha_\mathrm{max,sat} + C(\alpha_\mathrm{max,CofE} - \alpha_\mathrm{max,sat}), \tag{4}$$

where $C$ is a proportionality coefficient. We optimize two empirical parameters that show a positive relation to SA – $\alpha_\mathrm{max,CofE}$ and $C$, whose default values are 0.85 and 0.5, respectively (Fig. 1c–d): SA shows similar sensitivities to both parameters within the same range but is a bit more sensitive to $C$. Some other values have been used in previous studies (e.g., Livneh et al., 2010), such as 0.6 to 0.95 for $\alpha_\mathrm{max,CofE}$ and 1.0 for $C$. For the parameter estimation in this study, we set the ranges from 0.1 to 1.0 for both parameters.

### 2.1.3 Snow depth (SD)

In the Noah LSM, SD is evaluated as the ratio of SWE ($W_\mathrm{s}$) to snow density ($\mu_\mathrm{s}$), i.e., $W_\mathrm{s}/\mu_\mathrm{s}$ (Gotleib, 1980; Koren et al., 1999). While SWE is determined by precipitation in the model, snow density is determined by several other parameters such as the compression and melting of snow (Koren et al., 1999). Fresh snow density ($\mu_\mathrm{s,\,fresh}$) depends on air temperature ($T_\mathrm{air}$), i.e., 2 m temperature (Gotleib, 1980), as

$$\mu_\mathrm{s,\,fresh} = P_1 + P_2(T_\mathrm{air} + 15)^{1.5}, \tag{5}$$

where $P_1 = 0.05\,\mathrm{g\,cm^{-3}}$ and $P_2 = 0.0017\,\mathrm{g\,cm^{-3}\,^\circ C^{-1}}$ are the default values of the coefficients. If $T_\mathrm{air}$ is less than $-15\,^\circ\mathrm{C}$, $\mu_\mathrm{s,\,fresh}$ is set to $0.05\,\mathrm{g\,cm^{-3}}$; otherwise, $\mu_\mathrm{s,\,fresh}$ tends to increase as $T_\mathrm{air}$ increases. As the empirical parameters $P_1$ and $P_2$ are directly associated with $\mu_\mathrm{s,\,fresh}$, we seek

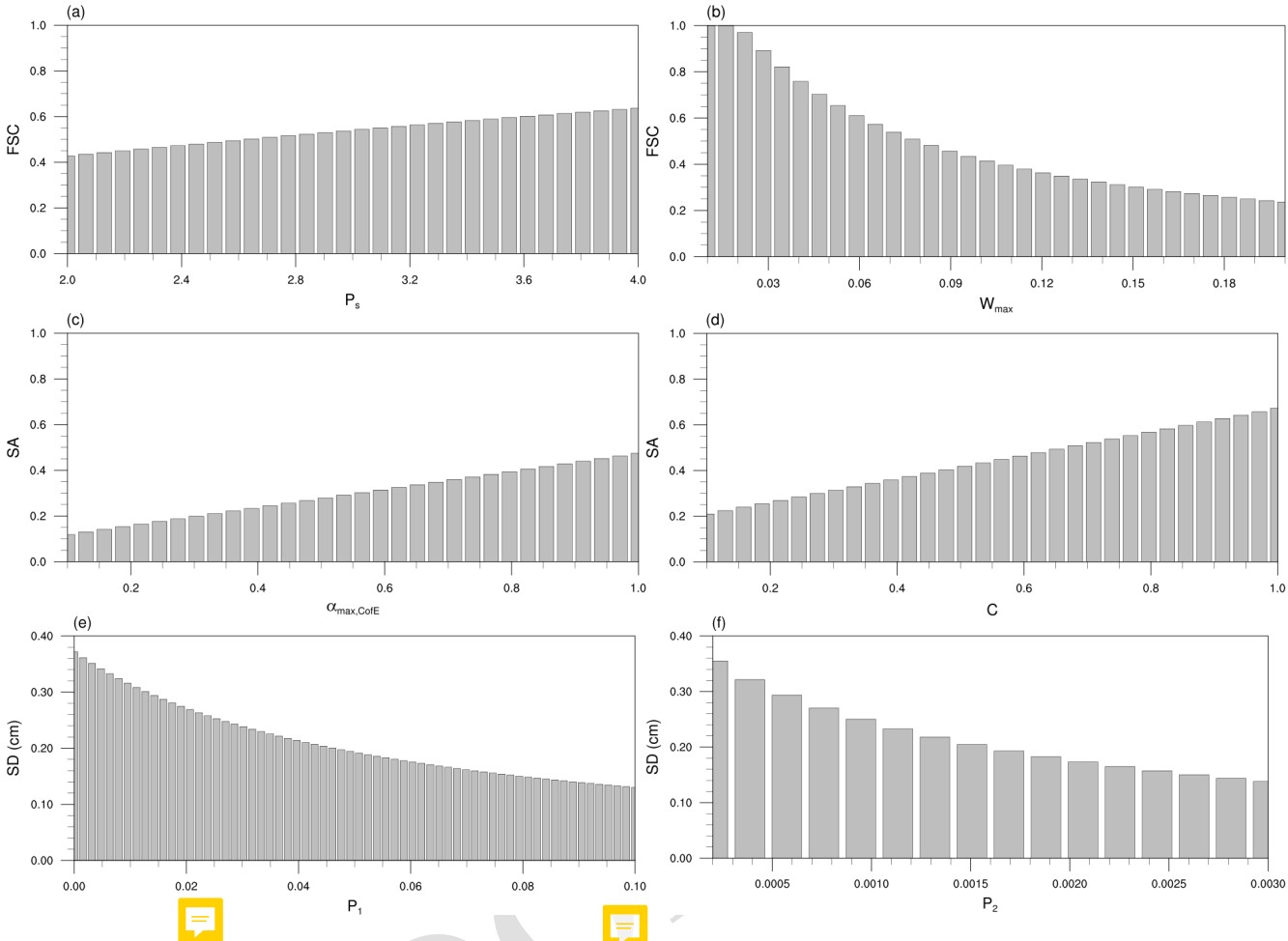

**Figure 1.** Responses of the snow variables to the variations in the snow-related parameters for given ranges. **(a, b)** Responses of FSC, for $W_s = 0.02$ TS1, to variations in $P_s$ (with $W_{max} = 0.08$ TS2) and in $W_{max}$ (with $P_s = 6$), respectively. **(c, d)** Responses of SA, for $\alpha_{max,sat} = 0.2$ and $t = 10$ d, to variations in $\alpha_{max,CofE}$ (with $C = 0.5$) and in $C$ (with $\alpha_{max,CofE} = 0.85$), respectively. **(e, f)** Responses of SD (cm), for $W_s = 0.02$ TS3 and $T_{air} = -5\,°C$, to variations in $P_1$ (with $P_2 = 0.0017$ TS4) and in $P_2$ (with $P_1 = 0.05\,\mathrm{g\,cm}^{-3}$).

optimal values of these parameters. Because snow density is inversely proportional to SD, both $P_1$ and $P_2$ have negative correlations with the SD (Fig. 1e–f), where the SD shows similar sensitivities to both parameters.

5 **2.2 Optimization tool: micro-genetic algorithm**

The genetic algorithm (GA) is a global optimization algorithm developed by John Holland in the 1970s (e.g., Holland, 1973, 1975) and is based on Darwinian principles of natural selection (Golberg, 1989). It uses reproduction selection, 10 crossover, and mutation to operate a set of potential solutions, i.e., *population* or *individuals*, which are expressed by a string, called a *chromosome*: its binary form is called a *gene* (Krishnakumar, 1990; Rudnaya and Santosa, 2000). The *selection* operator first selects good solutions or eliminates bad 15 solutions based on the fitness value; then, the *crossover* operator exchanges the genetic information between the solutions using the single-point or uniform types. The *mutation* operator modifies the value of each gene of the chromosomes by replacing it with the opposite value, e.g., 0 with 1, which prevents premature convergence. When a new generation is created, 20 the above processes are repeated until the convergence condition or the prescribed number of iterations is satisfied.

The micro-GA is an advanced and simplified GA with smaller generation sizes, thus requiring less computational time than the conventional GA (Krishnakumar, 1990; Wang 25 et al., 2010). It has been used in meteorology for optimal parameter estimation (e.g., Yu et al., 2013) or scheme-based optimization (e.g., Hong et al., 2014, 2015; Park and Park, 2021; Yoon et al., 2021). Its main difference from the conventional GA is the population size; for example, the micro-GA 30 uses 5 individuals, while the conventional GA uses more than 30 individuals. Note that the conventional GA with a small population quickly converges to non-optimal solutions due to insufficient information; however, the micro-GA solves this problem by using *elitism*, which assigns the best individual 35

among the five individuals based on the fitness evaluation and carries it to the next generation – this guarantees the preservation of the good solutions during the generations. Furthermore, the micro-GA does not take mutation to achieve diversity; instead, it uses the *re-initialization* which starts with a new individual whenever the diversity is lost.

### 2.2.1 Coupling the micro-GA with the Noah LSM and parallelization

Figure 2 describes the process of parameter optimization in the micro-GA–Noah LSM coupled system. (1) The micro-GA initializes the snow parameter combinations represented by the binary encoding through the random samples of the individual. (2) The micro-GA controls the Noah LSM by editing the parameter-related files, such as GEN-PARM.TBL, VEGPARM.TBL, and the Fortran code (module_sf_noahlsm.F), and prepares the forcing data for each station. (3) As recommended in Carroll (1996), the five individuals configured with the different snow parameters execute the ensemble runs of the Noah LSM in parallel. (4) The performance of each Noah LSM is evaluated in comparison with the observation through a given fitness function. (5) The micro-GA selects the highest fitness by comparing a number of individuals through the tournament selection. (6) New combinations for the next generation are produced through the crossover using the selected ones in the previous step. (7) When the convergence is satisfied, the other four individuals except the best individual marked by elitism are randomly regenerated. (8) The micro-GA repeats these processes until the prescribed entire iteration converges into a global maximum of the fitness function.

Although the micro-GA is computationally more efficient than the conventional GA, it still demands substantial computing time because each individual serially executes the model. Therefore, we have developed a parallel processing system in the micro-GA–Noah LSM coupled system. Instead of sequentially performing each individual and calculating the fitness within a generation, we run the model simultaneously for all populations to obtain the fitness and select the best individual when all tasks are finished (see the dashed box in Fig. 2). This new parallel system linearly reduces the execution time, which is proportional to the number of individuals. In addition, since the coupling system was created in a shell script, it is possible to assign multiple cores for model execution for various stations. The new parallel processing system, created by reflecting these two main points, improves the computation time – making it different from the non-parallel processing of a coupled system, e.g., the coupled micro-GA and the Noah LSM with multiple physics options (Noah-MP) model (see Hong et al., 2014).

### 2.2.2 Fitness function

The fitness function is a performance index to evaluate how well potential solutions fit the objective. In the GA optimization, the fitness function should be carefully defined because it is used for all generations and individuals. Generally, the root mean squared error (RMSE) is a widely used indicator for evaluating the performance of a model (e.g., Yan et al., 2019). Since our aim is to improve the snowfall prediction, we simultaneously evaluate all related snow variables – FSC, SA, and SD. We have first calculated the RMSE for each snow variable as

$$\text{RMSE}(\boldsymbol{x}) = \sqrt{\frac{\sum_{i=1}^{N} (\hat{\boldsymbol{x}}_i - \boldsymbol{x}_i)^2}{N}}, \tag{6}$$

where $\boldsymbol{x}$ is a vector representing the three snow variables and $N$ is the total number of observation times. Here, $\hat{\boldsymbol{x}}$ is the predicted values in the Noah LSM, while $\boldsymbol{x}$ is the observed values. The number of observations is dependent on the observational types: the Automated Synoptic Observing System (ASOS) produces hourly data for SD, while the MODerate resolution Imaging Spectroradiometer (MODIS), a sensor on board the polar-orbiting satellite Terra, produces daily data for FSC and SA. To calculate the RMSE between the model solutions and observations, the Noah LSM simulations are made over the observation locations. For SD, the RMSE is directly obtained on the same grid point. As the MODIS data have a coarser resolution, we use the observation point nearest the ASOS location (see the details in Sect. 2.3).

We then obtain the improvement ratio, $r(\boldsymbol{x})$, by comparing the RMSEs from the model runs with non-optimized parameters (say, CNTL) and optimized parameters (say, OPTM), respectively, as

$$r(\boldsymbol{x}) = \frac{\text{RMSE}(\boldsymbol{x})_{\text{CNTL}} - \text{RMSE}(\boldsymbol{x})_{\text{OPTM}}}{\text{RMSE}(\boldsymbol{x})_{\text{CNTL}}}. \tag{7}$$

Lastly, we have averaged all the improvement ratios for the snow variables to define the fitness function, $f(\boldsymbol{x})$, as

$$f(\boldsymbol{x}) = \sum_{j=1}^{M} \frac{r(\boldsymbol{x})_j q_j}{M}, \tag{8}$$

where $M$ is the number of stations and $q$ is a quality control flag (QCF) – either 0 or 1. The QCF is employed to secure a sufficient number of snow observations. It is set to 0 (i.e., the fitness function is not accumulated) for the following cases: (1) snow events are not simulated after optimization and (2) the number of snow observations is less than two. Furthermore, when the performance deteriorates after optimization, we give a penalty by doubling Eq. (7) to prevent degradation of the optimization.

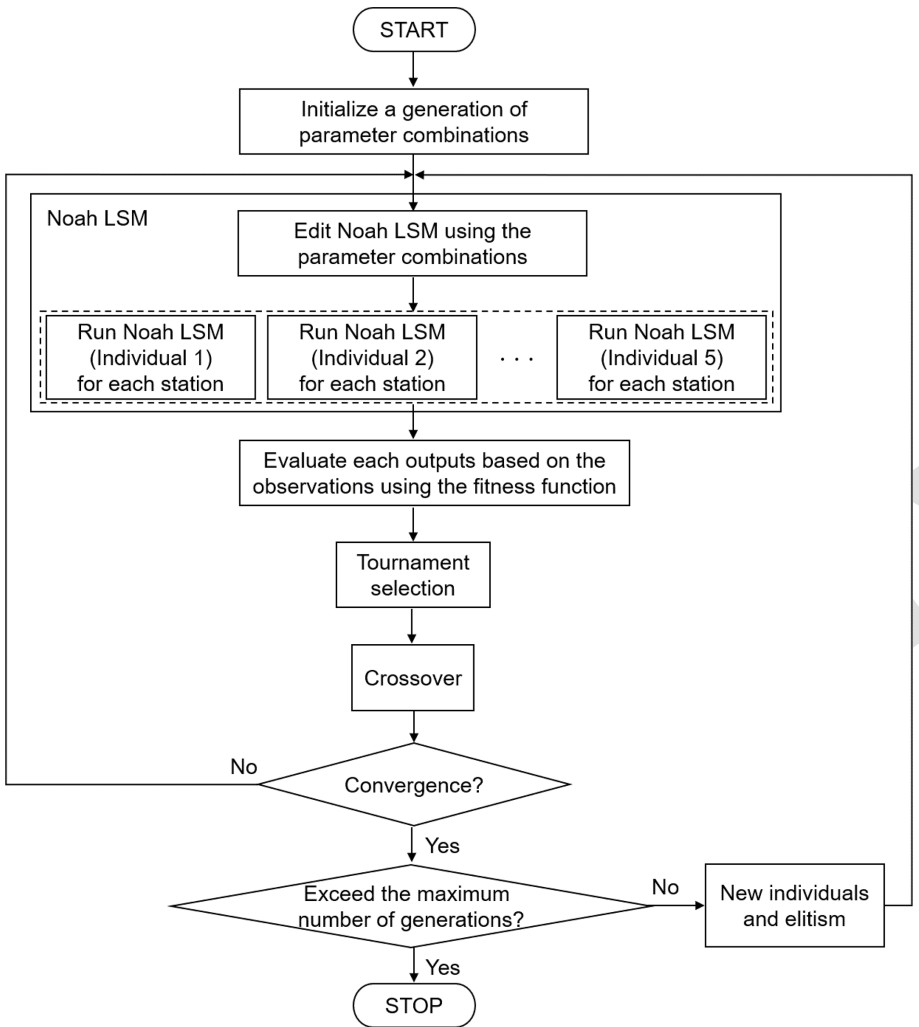

**Figure 2.** A flowchart of parameter optimization from the micro-GA–Noah LSM coupled system. The dashed box depicts the parallel system for the Noah LSM, running for each individual.

We finally define the normalized fitness function, $f_n(x)$, as

$$f_n(x) = \frac{f(\text{FSC}) + f(\text{SA}) + f(\text{SD})}{3}, \tag{9}$$

whose values lie in the range $[-1, 1]$. Thus, the micro-GA finds the maximum fitness based on Eq. (9).

## 2.3   Data

The land surface processes were forced by six meteorological fields from ASOS (https://data.kma.go.kr, last access: 24 October 2022): wind speed (m s$^{-1}$), wind direction (degrees), temperature (K), relative humidity (%), surface pressure (hPa), and precipitation rate (kg m$^{-2}$ s$^{-1}$). When missing data exist in less than 72 h, linear interpolation was performed except for precipitation. Stations with a missing rate greater than 1 %, during the entire experimental period, have been excluded. For the initial and boundary conditions,

downward shortwave/longwave radiation (W m$^{-2}$), precipitation rate (kg m$^{-2}$ s$^{-1}$), soil temperature (K), soil moisture (m$^3$ m$^{-3}$), and surface temperature (K) have been obtained from the European Centre for Medium-Range Weather Forecasts (ECMWF) – the fifth-generation ECMWF reanalysis-Land (ERA5L) hourly data (Muñoz-Sabater, 2019) – having a spatial resolution of 9 km and four soil layers with thicknesses of 7, 21, 72, and 189 cm, respectively, from top to bottom for a total depth of 2.89 m. We have used the data at the ERA5L grid's nearest point to the ASOS station.

The snow observations (i.e., SD, FSC, and SA) are used for the model verification and the fitness function calculation. For SD, the hourly model outputs are evaluated using the hourly ASOS data. To confirm the snow season, we have excluded the SD observations lower than 0.1 cm. For FSC and SA, we have no ASOS observations over SK; thus, we have used the MODIS/Terra Snow Cover Daily L3 Global 500 m SIN Grid radiance data (Hall and Riggs, 2021). They

are generated from the MODIS/Terra Snow Cover 5-Min L2 Swath 500 m data (Hall et al., 2006) by selecting the best observation based on a scoring algorithm when they are closest to nadir with maximum coverage of the cell (Hall and Riggs, 2007). In particular, FSC is generated by the normalized difference snow index (NDSI). The MODIS snow data at the points nearest to the ASOS locations were extracted and used for verification of the model-generated FSC and SA. Being a polar-orbiting satellite, MODIS contains only one observation per day; thus, we have extracted the model output for verification at 02:00 UTC, when the satellite (Terra) passes over SK. For the calculations, we have converted the percent values of FSC and SA to the decimal values; then, we excluded observational data with values below 0.05 (i.e., 5 %) for both FSC and SA.

For the optimization experiment, we have selected some stations that represent different land covers in SK, aiming at having a representative combination of snow-related parameters over SK. We have defined a representative set of LCTs within a 2.5 km radius from the ASOS observations, excluding the water body. The LCTs have been taken from the MODIS (on board Terra and Aqua) Land Cover Type Yearly Climate Modeling Grid (CMG) Version 6 (Friedl and Sulla-Menashe, 2015), in which maps are provided from the land cover classification schemes of the International Geosphere-Biosphere Programme (IGBP), the University of Maryland (UMD), and the leaf area index (LAI), all at a 0.05° spatial resolution in geographic latitude/longitude projection (see Sulla-Menashe and Friedl, 2018), for the entire globe from 2001 to 2019. Finally, we have compiled a set of five representative stations for each different LCT – deciduous broadleaf forest (DBF), mixed forest (MF), woody savanna (WS), cropland (CL), and urban and built-up lands (UB) – as shown in Table 1.

## 3 Experimental design

We have designed the following two GA optimization experiments: (1) OPT_5 that optimizes five snow parameters ($P_s$, $\alpha_{\max,\mathrm{CofE}}$, $C$, $P_1$, and $P_2$); (2) OPT_W that optimizes $W_{\max}$. These parameters are all constants and do not vary with time and space. Among the six parameters, only $W_{\max}$ depends on the LCTs, though it is still fixed for a given LCT. Thus, we conducted OPT_5 and OPT_W separately. Note that SK is represented by five different LCTs considering the sufficient days of snowfall and ASOS observation (see Table 1). Because OPT_5 optimizes with more parameters and generations, we have selected 10 stations (i.e., 2 stations per LCT) based on snowfall amount to reduce the computation time (see Fig. 3a). To investigate the performance of snow prediction through optimized snow parameters, we have designed the following three verification experiments for the 25 observation stations: (1) CNTL using non-optimized (i.e., default) parameters; (2) VRF_5 using the five optimized parameters

obtained from OPT_5; and (3) VRF_6 using the six optimized parameters obtained from both OPT_5 and OPT_W (see Fig. 3b).

For the micro-GA optimization, we have prespecified the following input parameters: (1) the population size, i.e., a collection of individuals; (2) the number of parameters to be used for optimization; (3) the number of chromosomes expressing an arbitrary solution; (4) the maximum number of generations to iterate the optimization; (5) the type of crossover operator that creates a new structure of chromosomes through the exchange of the chromosome; (6) the elitism to decide whether the most suitable individual would be preserved for the next generation. The micro-GA–Noah LSM coupled system has been repeatedly performed to find a parameter combination within the specified generations.

Table 2 describes the input parameters for the micro-GA used in this study. We follow the options known as the best performance in the micro-GA; this is done with a population size of five and a uniform crossover (i.e., crossover operator = 1.0) with elitism (Carroll, 1996; Yu et al., 2013; Yoon et al., 2021). The uniform crossover in which each gene is selected randomly from one of the parent chromosomes makes all populations perform a crossover at every generation to acquire diversity (Lee et al., 2005). The number of parameters to be optimized is five for OPT_5 and one for OPT_W. The number of chromosomes determines the number of cases expressed in a binary format. For example, the selected parameters – $P_s$, $\alpha_{\max,\mathrm{CofE}}$, $C$, $P_1$, $P_2$, and $W_{\max}$ – use different chromosomes, i.e., 5, 5, 5, 6, 4, and 5, respectively; thus, the total number of chromosomes is 30 for OPT_5 and 5 for OPT_W. The maximum value of generations at the end of optimization is generally set to 100 (Yu et al., 2013; Yoon et al., 2021; Zhu et al., 2019), whereas we increased generations up to 200 in OPT_5 due to the larger number of parameters to be optimized.

In this study, we have conducted the optimization experiments from 00:00 UTC 1 May 2009 to 23:00 UTC 30 April 2018. During this 9-year period, the number of snow observations was continuously secured. Data from the first 5 months (May–October in 2009) were utilized for model initialization and spinup, and thus they were not considered for the verification. Cross-validation has been conducted using the 1-year data from 00:00 UTC 1 May 2018 to 23:00 UTC 30 April 2019. Since they showed similar aspects, we only discuss the results of optimization periods with sufficient samples.

## 4 Results

### 4.1 Spinup analysis

Numerical prediction models generally require spinup to reach a statistical equilibrium state where the initial conditions under a forcing are adjusted to the model's own physic-

**Table 1.** Five representative LCTs over SK, following the IGBP classification – DBF, MF, WS, CL, and UB. For each LCT, five selected stations are shown with the station name (abbreviation in parentheses), location in latitude (° N) and longitude (° E), ratio of LCT in a 2.5 km buffer (%), soil type, and missing ratio (%). The OPT_5 experiment employs only the stations highlighted in bold, while the other experiments use all the stations.

| IGBP LCT | Station name | Latitude | Longitude | Ratio of LCT in the 2.5 km buffer | Soil type | Missing ratio |
|---|---|---|---|---|---|---|
| DBF | **Ulleungdo (UL)** | **37.481** | **130.899** | **82.7** | **Silt loam** | **0.15** |
| | Taebaek (TB) | 37.170 | 128.989 | 67.0 | Loam | 0.15 |
| | Inje (IJ) | 38.060 | 128.167 | 62.7 | Sandy loam | 0.07 |
| | Chupungnyeong (CP) | 36.220 | 127.995 | 56.8 | Silt loam | 0.04 |
| | **Youngwol (YW)** | **37.181** | **128.457** | **42.6** | **Clay** | **0.09** |
| MF | Bongwha (BW) | 36.944 | 128.914 | 38.7 | Loam | 0.11 |
| | Hapcheon (HP) | 35.565 | 128.170 | 32.1 | Loam | 0.51 |
| | Hongcheon (HC) | 37.683 | 127.880 | 26.3 | Silty clay loam | 0.05 |
| | **Miryang (MY)** | **35.491** | **128.744** | **22.5** | **Sandy loam** | **0.16** |
| | **Gumi (GM)** | **36.131** | **128.321** | **24.1** | **Sandy loam** | **0.05** |
| WS | Imsil (IS) | 35.612 | 127.286 | 53.1 | Sandy loam | 0.12 |
| | Andong (AD) | 36.573 | 128.707 | 43.9 | Loamy sand | 0.04 |
| | Boeun (BE) | 36.488 | 127.734 | 41.2 | Sandy loam | 0.07 |
| | **Uljin (UJ)** | **36.992** | **129.413** | **39.2** | **Loam** | **0.19** |
| | **Bukgangneong (NG)** | **37.805** | **128.855** | **37.5** | **Sandy loam** | **0.04** |
| CL | Buan (BA) | 35.730 | 126.717 | 87.8 | Loam | 0.03 |
| | Icheon (IN) | 37.264 | 127.484 | 74.6 | Sandy loam | 0.16 |
| | Haenam (HN) | 34.554 | 126.569 | 63.7 | Sandy loam | 0.29 |
| | **Boryeong (BR)** | **36.327** | **126.557** | **53.8** | **Silty clay loam** | **0.14** |
| | **Jeongeup (JE)** | **35.563** | **126.839** | **51.7** | **Silt loam** | **0.28** |
| UB | Gwangju (GJ) | 35.173 | 126.892 | 94.6 | Loam | 0.03 |
| | **Seoul (SL)** | **37.571** | **126.966** | **90.8** | **Loam** | **0.08** |
| | Daejeon (DJ) | 36.372 | 127.372 | 72.2 | Sandy loam | 0.03 |
| | **Suwon (SW)** | **37.257** | **126.983** | **71.4** | **Sandy loam** | **0.10** |
| | Incheon (IC) | 37.478 | 126.625 | 70.1 | Loam | 0.07 |

**Table 2.** The input parameters for micro-GA in experiments OPT_5 and OPT_W.

| Input parameter | OPT_5 | OPT_W |
|---|---|---|
| Population size | 5 | 5 |
| Crossover operator | 1.0 | 1.0 |
| Elitism | On | On |
| Number of parameters | 5 | 1 |
| Number of chromosomes | 30 | 5 |
| Maximum value of generations | 200 | 100 |

s/dynamics and numerics (Bonekamp et al., 2018). Without sufficient spinup, the LSMs can generate severe bias of initial conditions (Cosgrove et al., 2003). Prior to the optimization experiments, we have conducted a spinup experiment in one of the stations, Seoul, to check the appropriate spinup time. It was carried out in two ways: (1) using a spinup period recursive in 9 years (e.g., Jun et al., 2020) and (2) using a spinup period that was not included in the analysis.

First, the Noah LSM has been repeatedly executed using the atmospheric forcing for 9 years. This recursive simulation has been conducted from 1 May 2009 to 30 April 2018 to see whether the model was able to reach an equilibrium by setting the repetition loop to 0, 300, 600, and 1000. Our results indicated no significant differences; thus, we concluded that repetition was not required. Second, we have performed sensitivity tests to identify the spinup period due to changes in the initial conditions by adding biases ($\pm 0.1\,\mathrm{m^3\,m^{-3}}$ for soil moisture and $\pm 3\,\mathrm{K}$ for soil temperature) to the ERA5L data. As a result, we found that the adequate spinup periods were about 3 months and 1 year for soil moisture and soil temperature, respectively; however, the snow variables were insensitive to the initial condition changes, thus requiring no spinup period. Although the spinup is not necessary for this study that focuses on the snow processes, we have performed the optimization experiments starting from May when snow is absent.

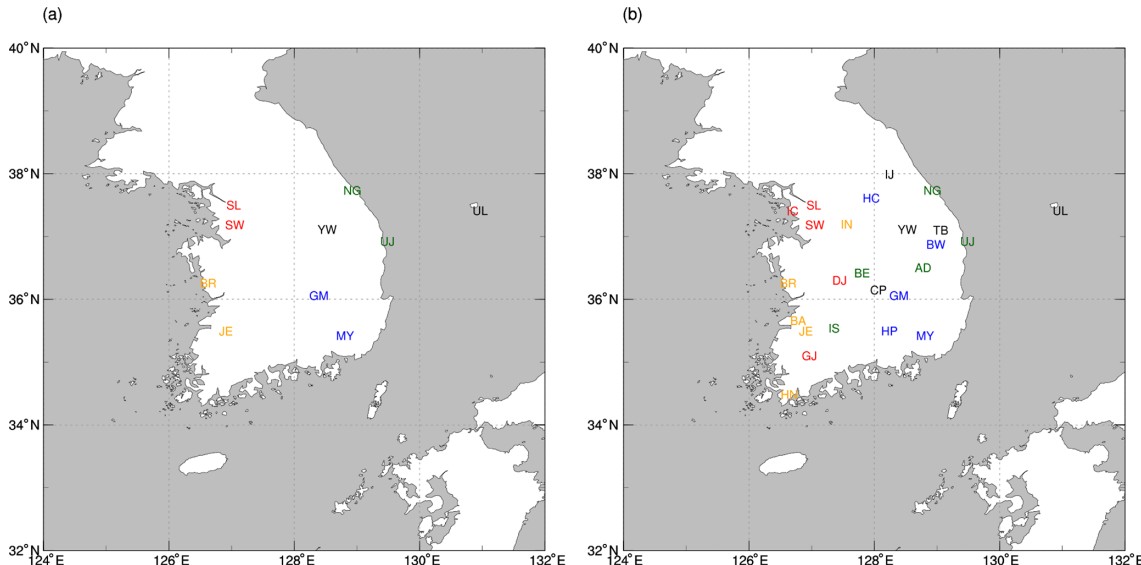

**Figure 3.** Stations used for experiments **(a)** OPT_5 and **(b)** OPT_W, CNTL, VRF_5, and VRF_6. Different colors in the station abbreviations represent different LCTs: DBF (black), MF (blue), WS (green), CL (yellow), and UB (red). See Table 1 for the abbreviations of the stations and LCTs.

## 4.2 Optimal estimation of snow parameters

To optimize snow parameters specialized in SK, we have employed the micro-GA–Noah LSM coupled system using the observations over SK. Figure 4a shows the evolution of the fitness function for OPT_5 in a total of 200 generations and Fig. 4b for OPT_W in a total of 100 generations. Since OPT_W optimizes solely the $W_{max}$ parameter, it has smaller generations. In OPT_5, the fitness function converges at the 160th generation, while the fitness function of OPT_W quickly converges in all LCTs (Fig. 4b). The convergence occurs at the 3rd generation for DBF, 70th generation for MF, 7th generation for both WS and CL, and 12th generation for UB.

As a result, we have obtained the optimized six snow parameters over SK (Table 3). OPT_5 simultaneously generates the optimized five snow parameters ($P_s$, $\alpha_{max,CofE}$, $C$, $P_1$, and $P_2$) associated with the FSC, SA, and SD, while OPT_W, depending on the LCTs, generates the optimized $W_{max}$ associated with the FSC. The first snow parameter, $P_s$, is optimized from its standard value of 2.6 to 2.7097, which results in an increase in the FSC. The second snow parameter, $W_{max}$, is optimized depending on each LCT. In detail, the $W_{max}$ in DBF and WS increases from 0.08 to 0.1632[TS5] and from 0.03 to 0.0406[TS6], respectively. They lead to a decrease in the FSC due to a negative correlation. On the other hand, the $W_{max}$ in MF and UB decreases from 0.08 to 0.0529[TS7] and from 0.04 to 0.0284[TS8], respectively, thus increasing the FSC. The optimized CL shows a similar value from 0.04 to 0.0406[TS9], which means that the current value was correct for SK. The third snow parameter related to the SA, $\alpha_{max,CofE}$, decreases from 0.85 to 0.7387, inducing a decrease in SA. The fourth snow parameter, $C$, also shows a similar value from 0 to 0.5355, and thus this value was correct for SK. The fifth snow parameter, $P_1$, increases from 0.05 to 0.0698[TS10], resulting in a decrease in SD. The last snow parameter, $P_2$, decreases from 0.0017 to 0.0002[TS11], leading to an increase in SD.

We have investigated the mean bias (MB) using the box plot expressing the quartile and the distribution of extreme values: it explains how much the bias of the CNTL is reduced in optimization experiments by comparing the model with the observations. Before optimization, the CNTL showed underestimated FSC and SD and overestimated SA ($-0.133$, $-4.39$ cm, and 0.0408, respectively; see Fig. 5). However, the bias patterns in FSC and SA vary for each station owing to the lower spatial and temporal resolution of satellite observations. On the other hand, the SD shows an underestimation at all stations; the increase in the SD due to fresh snow was underestimated, and snowmelt was proceeding faster than the observation.

The performance has been evaluated using the improvement ratio, which indicates how much the RMSE, MB, and coefficient of determination ($R^2$) of experiments using optimized parameters (i.e., VRF_5 and VRF_6) are improved compared to CNTL, as shown in Eq. (7) (Table 4). In VRF_5, new parameter values – $P_s$, $\alpha_{max,CofE}$, $C$, $P_1$, and $P_2$ – optimized by the micro-GA result in an improvement in the RMSEs for FSC, SA, and SD of 0.7 %, 5.4 %, and 13.7 %, respectively. However, the RMSE of FSC relatively weakly improved by about 0.7 % because the other parameter, $W_{max}$, is not yet optimized. In terms of MB, we anticipate that the increase in $P_s$ will overcome the underestimated FSC. However, VRF_5 strengthens the underestimation of FSC from

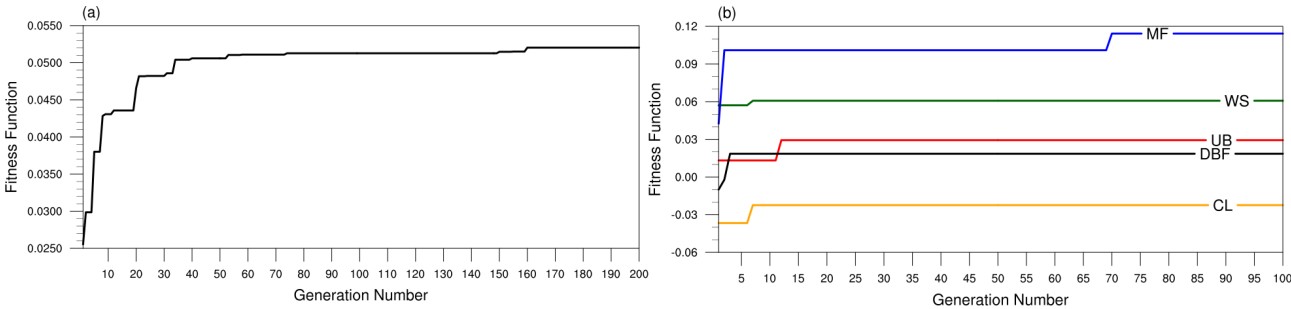

**Figure 4.** The fitness function for generations during **(a)** five-snow-parameter optimization (OPT_5) and **(b)** $W_{max}$ optimization (OPT_W) for DBF (black), MF (blue), WS (green), CL (yellow), and UB (red) LCTs.

**Table 3.** Summary of optimized snow parameters related to snow variables. Minimum (min), default, and maximum (max) are the ranges used in the optimization process. Default is the empirical value used in the Noah LSM.

| Snow variable | Snow parameter | LCTs | Min/default/max | Optimized value |
|---|---|---|---|---|
| FSC | $P_s$ | – | 2.0/2.6/4.0 | 2.7097 |
| | $W_{max}$ | DBF | 0.01/0.08/2.00 | 0.1632 |
| | | MF | 0.01/0.08/2.00 | 0.0529 |
| | | WS | 0.01/0.03/2.00 | 0.0406 |
| | | CL | 0.01/0.04/2.00 | 0.0406 |
| | | UB | 0.01/0.04/2.00 | 0.0284 |
| SA | $\alpha_{max,CofE}$ | – | 0.10/0.85/1.00 | 0.7387 |
| | $C$ | – | 0.1/0.5/1.0 | 0.5355 |
| SD | $P_1$ | – | 0.00/0.05/0.10 | 0.0698 |
| | $P_2$ | – | 0.0002/0.0017/0.003 | 0.0002 |

−0.133 to −0.145, and thus it deteriorates the MB by about 9.1 % (Table 4 and Fig. 5a). Regarding the SA, the optimized $\alpha_{max,CofE}$ decreases the SA to solve the overestimation in CNTL. The other parameter, $C$, has been optimized to its default value, 0.5355, which means that this was an appropriate constant for SK snowfall prediction. Therefore, the MB of the SA is improved by 26.9 % by reducing the SA from 0.0408 to 0.0298 (Table 4 and Fig. 5b). Next, SD shows the greatest RMSE improvement of 13.7 %. In fact, the Noah LSM suffers from a negative bias for SWE, especially in early spring (Sheffield et al., 2003; Ek et al., 2003; Pan et al., 2003; Mitchell et al., 2004; Jin and Miller, 2007; Livneh et al., 2010). Because SD is proportional to SWE, the underestimation can be exhibited due to negative bias of the SWE. However, the optimized $P_1$ leads to a decrease in SD, and thus it intensifies the underestimation for SD. On the other hand, the optimized $P_2$ increases the SD as follows: when the air temperature is warmer than −15 °C, the fresh snow density slowly increases, which quickly induces an increase in SD following Eq. (5). Therefore, the optimization of $P_2$ solves the underestimated SD by about 35.9 % due to the increased SD from −4.39 to −2.81 cm within most of the temperature ranges (Table 4 and Fig. 5c). We also investigated $R^2$, which measures the proportion of variation for a dependent variable that can be explained by an independent variable. Although the $R^2$ values are low in FSC and SA, the difference between CNTL and the verification experiment (e.g., VRF_5) has 95 % statistical significance, as evaluated with a two-tailed $t$ test. After optimization, the $R^2$ values in VRF_5 improve by 3.3 % and 1.5 % for FSC and SD, respectively. However, these changes are insignificant compared to the other statistics such as RMSE and MB.

To supplement insufficient improvement in the FSC, we have additionally optimized $W_{max}$ as a function of LCT (OPT_W) using the optimized values of five parameters from OPT_5. Here, we have only used the FSC to define the fitness function, not considering SA and SD; thus, the fitness function is defined using Eq. (8), where the FSC is the only element of $x$, and the normalized process with Eq. (9) is not necessary. As a result, OPT_W further improves the RMSE of the FSC in VRF_6 compared to VRF_5 in most stations: the significant decreases in $W_{max}$ over MF and UB lead to an increase in the FSC, possibly alleviating the underestimation problem of the FSC in VRF_5.

Finally, all six parameters related to the snow variables have been verified in VRF_6 with the same 25 stations used in CNTL. When the optimized five parameters are used except $W_{max}$ (VRF_5), SA and SD are improved, and FSC shows a weak improvement in RMSE performance (Table 4). However, when the optimized $W_{max}$ depending on the LCTs

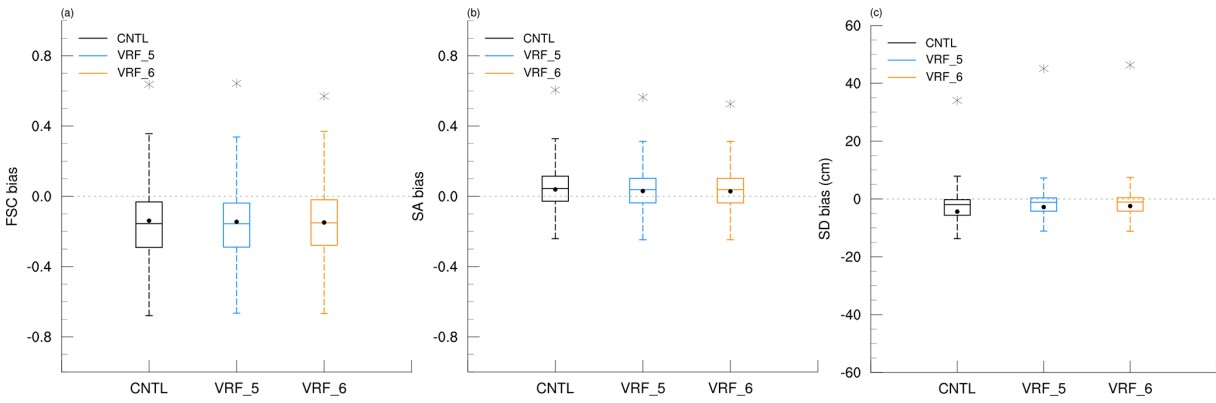

**Figure 5.** Box plots of **(a)** FSC bias, **(b)** SA bias, and **(c)** SD bias (cm) for CNTL, VRF_5, and VRF_6. The maximum differences are indicated with the black star symbol (e.g., 0.637 (CNTL), 0.643 (VRF_5), 0.570 (VRF_6) for FSC, 0.605 (CNTL), 0.563 (VRF_5), and 0.525 (VRF_6) for SA and 34.1 cm (CNTL), 45.1 cm (VRF_5), and 46.3 cm (VRF_6) for SD). Each mean of snow variables is indicated as a black circle (e.g., −0.133 (CNTL), −0.145 (OPT_5), and −0.149 (VRF_6) for FSC, 0.0408 (CNTL), 0.0298 (VRF_5), and 0.0281 (VRF_6) for SA and −4.39 cm (CNTL), −2.81 cm (VRF_5), and −2.45 cm (VRF_6) for SD).

**Table 4.** The RMSE, MB, and $R^2$ of snow variables and improvement ratios (%) in parentheses from CNTL to VRF_5, and VRF_6 over the 25 representative stations. The difference between CNTL and verification experiments (i.e., VRF_5 and VRF_6) has 95 % statistical significance, as evaluated with a two-tailed $t$ test.

| Experiments | CNTL | | | VRF_5 | | | VRF_6 | | |
|---|---|---|---|---|---|---|---|---|---|
| Snow variable | FSC | SA | SD | FSC | SA | SD | FSC | SA | SD |
| RMSE | 0.249 | 0.132 | 9.094 | 0.247 (0.7 %) | 0.125 (5.4 %) | 7.847 (13.7 %) | 0.124 (3.3 %) | 0.125 (6.2 %) | 7.547 (17.0 %) |
| MB | −0.133 | 0.0408 | −4.39 | −0.145 (−9.1 %) | 0.0298 (26.9 %) | −2.81 (35.9 %) | −0.149 (−11.9 %) | 0.0281 (31.0 %) | −2.45 (44.2 %) |
| $R^2$ | 0.257 | 0.281 | 0.808 | 0.265 (3.3 %) | 0.276 (−1.7 %) | 0.821 (1.5 %) | 0.277 (8.0 %) | 0.274 (−2.2 %) | 0.834 (3.0 %) |

from OPT_W is used (VRF_6), the FSC appears in a larger positive impact with other variables. As a result, an improvement in RMSE for FSC, SA, and SD is 3.3 %, 6.2 %, and 17.0 %, respectively. However, the MB for the FSC strengthens from 9.1 % to 11.9 % in VRF_6 (Table 4 and Fig. 5a) due to larger negative bias, especially in the DBF. On the other hand, SA and SD reduce the MB against the CNTL and enhance the improvement ratio from 26.9 % to 31.0 % and from 35.9 % to 44.2 %, respectively (Table 4 and Fig. 5b–c). Like the RMSE, the $R^2$ of FSC and SD also improved in VRF_5 and VRF_6. The SA worsened in VRF_5 and was a bit more severe in VRF_6. However, they are still small impacts compared to RMSE and MB.

To understand more details of the improvements due to the optimization, we analyzed the scatter plots that compare the observations and the model results in Fig. 6 and listed their RMSE and $R^2$ in Table 5. Since the observation patterns are different for different stations, we selected the representative station for each LCT. For FSC, it is relatively hard to recognize the explicit bias patterns, as shown in Fig. 6

(left panels); however, compared to CNTL, the RMSE decreased in VRF_5 and further reduced in VRF_6 (see Table 5). The VRF_6 revealed the largest $R^2$ values over most LCTs, except WS (station NG) and CL (station BR). In particular, VRF_6 produced the highest FSC over MF (station GM) (see Fig. 6d), with the smallest RMSE and the largest $R^2$, which significantly alleviated the underestimation problem. For SA, its overestimation in CNTL has been prominently reduced in both VRF_5 and VRF_6 – see Fig. 6 (middle panels). For instance, SA decreased over DBF (station UL) in both VRF_5 and VRF_6, with a larger decrease for VRF_6 (Fig. 6b). The performance statistics of both VRF_5 and VRF_6 demonstrated improvements over most LCTs except UB (station SL) (see Table 5). For SD, the parameter optimization brought about remarkable improvement compared to FSC and SA – see Fig. 6 (right panels). Note that SD is optimized using the hourly in situ observations (i.e., larger amount of data), while both FSC and SA are optimized using the daily satellite observations. For example, VRF_6 with DBF produced notably large SD values (Fig. 6c) with the

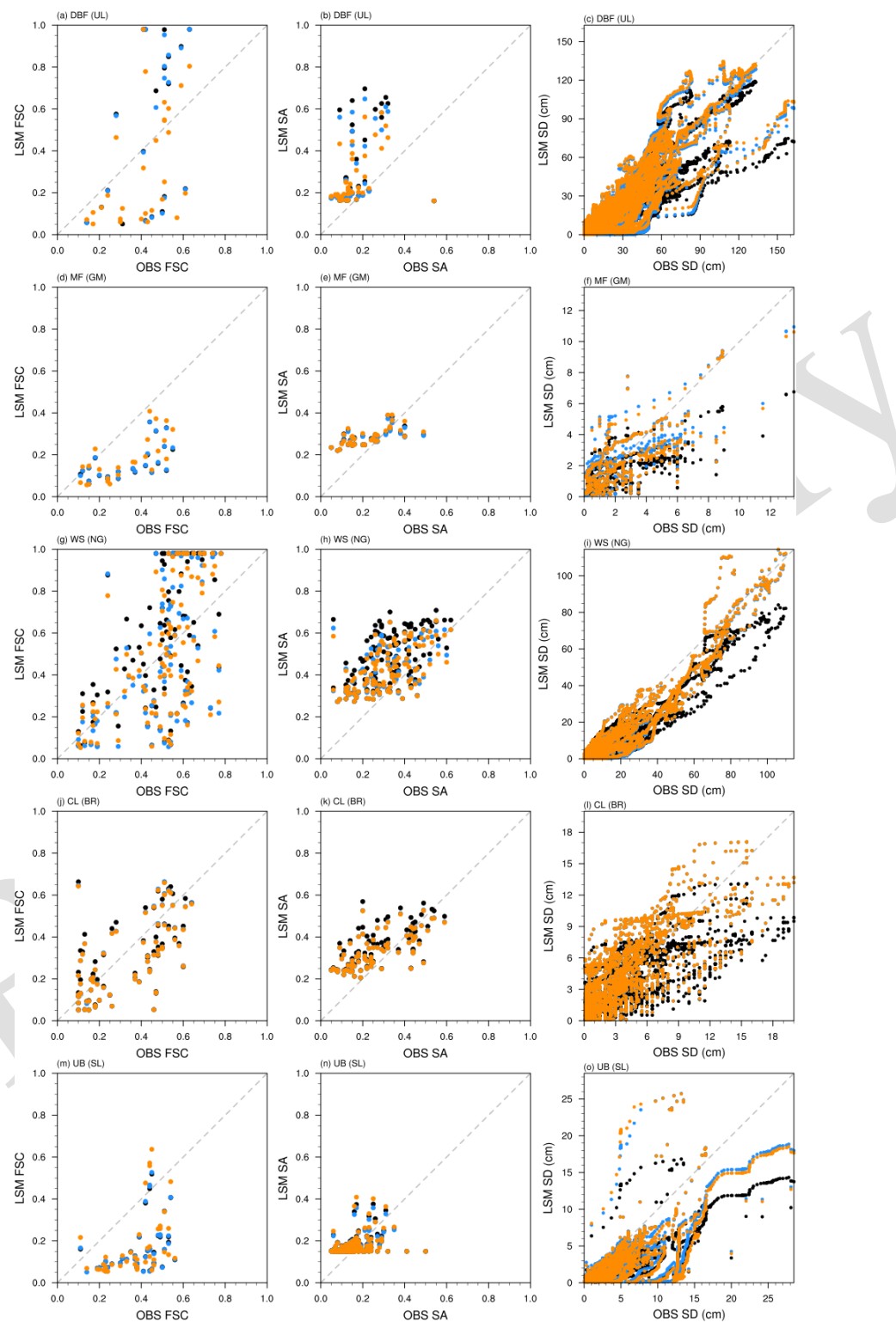

**Figure 6.** Scatter plots of observations (OBS) and model results (LSM) for snow variables FSC (left panels), SA (middle panels), and SD (cm; right panels) from the verification experiments – CNTL (black dots), VRF_5 (blue dots), and VRF_6 (orange dots), which are evaluated over different LCTs. **(a–c)** DBF represented by station UL, **(d–f)** MF by GM, **(g–i)** WS by NG, **(j–l)** CL by BR, and **(m–o)** UB by SL.

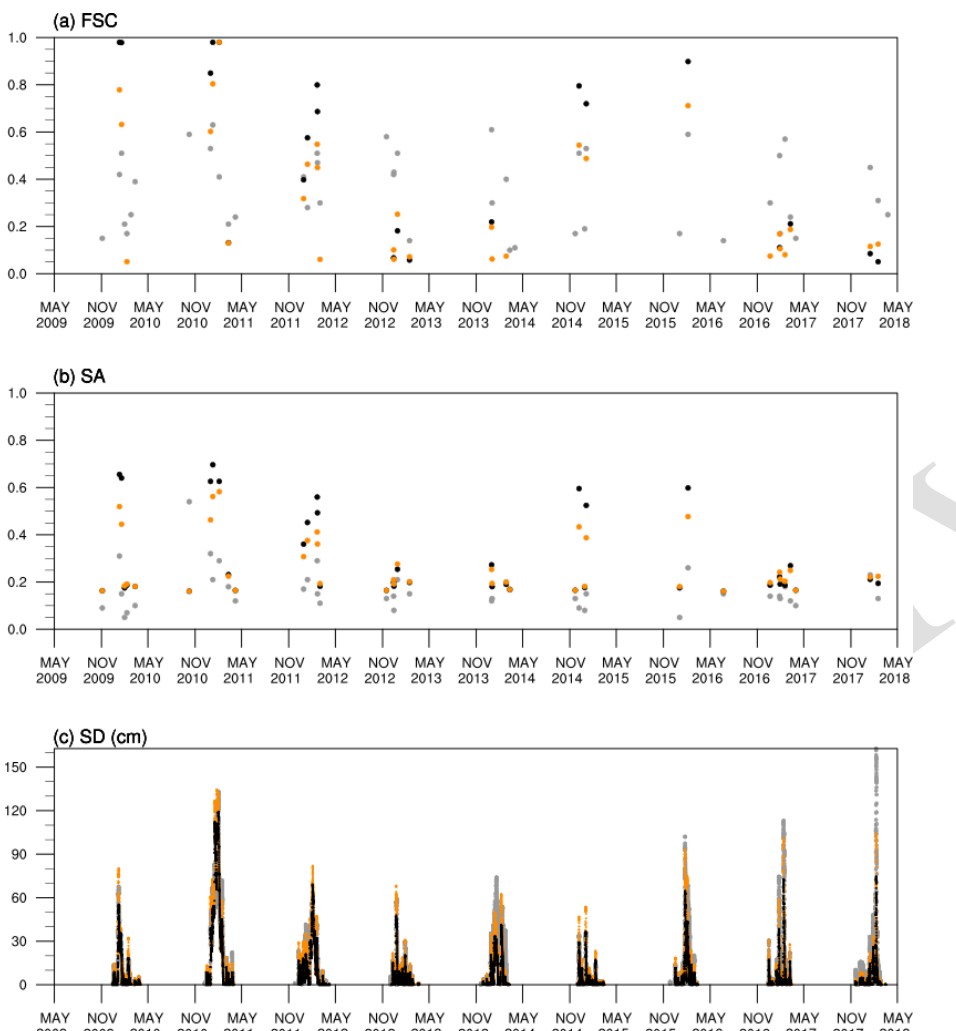

**Figure 7.** Time series of the snow variables for DBF (e.g., UL) from May 2009 to April 2018: **(a)** FSC, **(b)** SA, and **(c)** SD (cm). Observations are in gray dots, and model results are in black dots for CNTL and in orange dots for VRF_6.

lowest RMSE and the highest $R^2$ (Table 5), diminishing the underestimation problem in CNTL. It is hard to say which verification experiment gives the best results (i.e., VRF_5 versus VRF_6), but the performance with optimized parameters is usually better than CNTL in terms of RMSE (e.g., for most LCTs such as DBF, MF, WS, and UB) and $R^2$ (e.g., for LCTs including DBF, MF, and CL). Overall, both VRF_5 and VRF_6 produced snow variables that are closer to observations than CNTL for most LCTs (i.e., stations), and VRF_6 generally showed the lowest RMSE and the highest $R^2$ in all the snow variables.

Figure 7 shows temporal changes in the snow variables after parameter optimization by comparing their time series of the observations and the model simulations (CNTL versus VRF_6) for DBF represented by UL. The CNTL shows positive or negative biases in FSC, positive bias (overestimation) in SA, and negative bias (underestimation) in SD: these

biases are all reduced down in VRF_6. The bias patterns in Fig. 7 are consistent with those in Fig. 6a–c.

Lastly, we have investigated how the optimized snow parameters can affect the other variables in the LSM. Figure 8 depicts the time series of the differences of LSM variables (soil temperature, sensible heat flux, and soil moisture) between VRF_6 and CNTL (i.e., VRF_6 minus CNTL) following the changes in SD. Although the LSM variables here are not directly optimized, they respond to the optimized snow parameters through associated physical processes. Note that the underestimation of SD in CNTL has been alleviated in VRF_6 by using the optimized snow parameters (see Figs. 7c and 8a). Next, soil temperature in the first soil layer (7 cm) increases as SD increases after optimization, which consequently increases sensible heat flux. The residual of the surface energy balance is close to zero, implying that the surface energy balance is well conserved even after optimization. Soil moisture depends on snowmelt, following the trend

**Table 5.** Statistics of model performance using non-optimized parameters (CNTL) and optimized parameters (VRF_5 and VRF_6) over different LCTs represented by different stations – DBF represented by UL, MF by GM, WS by NG, CL by BR, and UB by SL. The RMSEs and $R^2$ values are shown for three snow variables – FSC, SA, and SD.

| Statistics | | RMSE | | | $R^2$ | | |
| --- | --- | --- | --- | --- | --- | --- | --- |
| LCT | Snow variable | CNTL | VRF_5 | VRF_6 | CNTL | VRF_5 | VRF_6 |
| DBF (UL) | FSC | 0.328 | 0.327 | 0.252 | 0.248 | 0.215 | 0.256 |
| | SA | 0.218 | 0.197 | 0.159 | 0.157 | 0.157 | 0.176 |
| | SD | 15.763 | 13.640 | 12.616 | 0.764 | 0.781 | 0.796 |
| MF (GM) | FSC | 0.208 | 0.206 | 0.178 | 0.388 | 0.408 | 0.520 |
| | SA | 0.105 | 0.103 | 0.103 | 0.411 | 0.421 | 0.460 |
| | SD | 1.789 | 1.526 | 1.542 | 0.435 | 0.502 | 0.493 |
| WS (NG) | FSC | 0.279 | 0.269 | 0.249 | 0.354 | 0.333 | 0.341 |
| | SA | 0.196 | 0.160 | 0.156 | 0.314 | 0.328 | 0.324 |
| | SD | 9.836 | 8.231 | 8.009 | 0.895 | 0.887 | 0.888 |
| CL (BR) | FSC | 0.163 | 0.160 | 0.160 | 0.363 | 0.385 | 0.384 |
| | SA | 0.132 | 0.122 | 0.122 | 0.443 | 0.457 | 0.456 |
| | SD | 2.542 | 2.583 | 2.590 | 0.478 | 0.540 | 0.539 |
| UB (SL) | FSC | 0.255 | 0.252 | 0.242 | 0.184 | 0.195 | 0.195 |
| | SA | 0.071 | 0.070 | 0.073 | 0.150 | 0.148 | 0.124 |
| | SD | 4.790 | 4.286 | 4.699 | 0.484 | 0.449 | 0.385 |

of increased snowfall in the previous winter. Extreme fluctuations sometimes appear in the time series analyses due to nonlinear effects, but we can understand the overall tendency according to the increased SD on the land surface.

## 5 Conclusions and outlook

The Noah land surface model (Noah LSM) generally underestimates snow amount during the peak winter and shows earlier snowmelt in spring, whereas it overestimates snow albedo (SA) over Eurasia, mainly due to uncertain parameterization processes (Saha et al., 2017). Our experiment with no optimization (CNTL) reveals underestimation of snow depth (SD) and fractional snow cover (FSC) and overestimation of SA compared to the in situ or satellite observations. Therefore, we have developed a coupled system of the micro-genetic algorithm (micro-GA) and the Noah LSM to reduce the uncertainties in parameterized snow processes through optimization of parameter values. This parameter estimation is an effort to further improve the model performance by reducing uncertainty in pre-existing parameterization schemes by optimizing the parameter values inside the schemes based on the observational data that reflect local characteristics to improve snow simulation. If the employed parameterization scheme has less uncertainty, improvement by parameter estimation in that scheme may not be significant; if the scheme has large uncertainty in parameter values, parameter estimation may bring about prominent improvement in the scheme's performance.

The coupling system of the micro-GA and Noah LSM automatically estimates the optimal snow-related parameters by objectively comparing observations and model solutions through the fitness function. Instead of trial-and-error procedures, it has the advantage of reducing a substantial amount of computational time. The original micro-GA reduces the computational time using the elitism and re-initialization methods in the small number of individuals. However, we have developed a parallel system on the coupled system to further improve the computational efficiency in this study; it enables us to simultaneously execute multiple individuals in one generation and multiple Noah LSM runs in one individual.

Six parameters included in the snow processes in the Noah LSM have been optimized by using a micro-GA during the period 2009–2018 in South Korea (SK). The first parameter is the distribution shape parameter that participates in the FSC calculation and shows a positive correlation with the FSC: the optimized value is expected to increase the FSC, but it is not sufficient to alleviate its underestimation problems. The second parameter is the snow water equivalent threshold value that implies 100 % snow cover and is also used in the FSC calculation depending on the land cover type: its optimized value improves the FSC in terms of RMSE and mean bias over some stations. The third parameter is the maximum SA coefficient: its optimized (decreased) value improves the RMSE by reducing the overestimation of SA. The fourth parameter is the coefficient in the maximum albedo of fresh snow, and its optimized value was similar to the default one. The other two parameters are related to the fresh snow

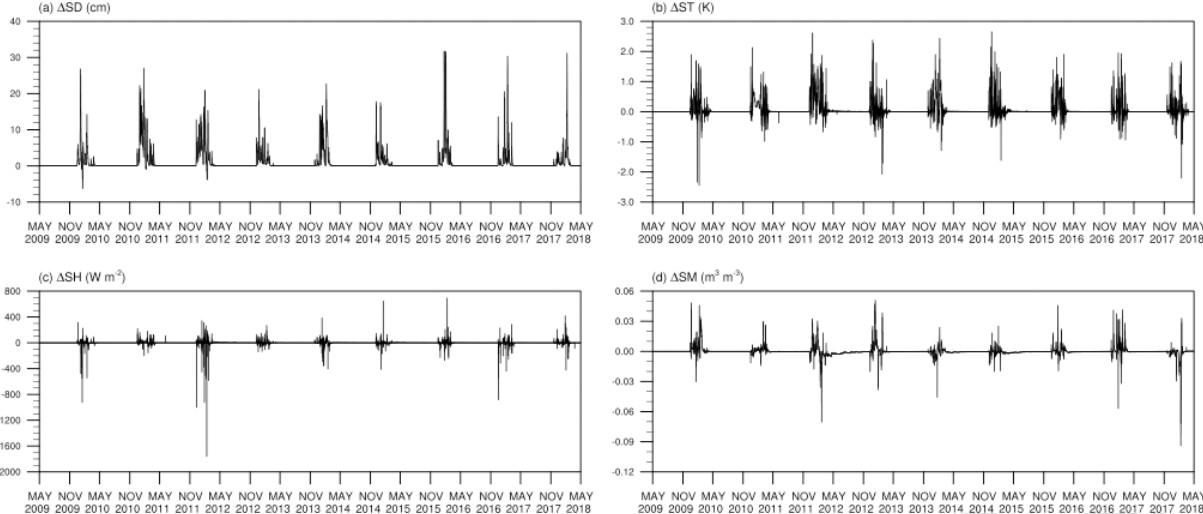

**Figure 8.** Time series of the difference between CNTL and VRF_6 for the UL in DBF from May 2009 to April 2018: **(a)** SD (cm), **(b)** soil temperature in the top soil layer (7 cm) (ST; K), **(c)** sensible heat flux (SH; W m$^{-2}$), and **(d)** soil moisture in the top soil layer (7 cm) (SM; m$^3$ m$^{-3}$).

density used for the SD calculation. In particular, the sixth parameter depends on air temperature, and its optimization brings about the largest improvement in SD: the optimized (reduced) value remarkably reduces the RMSE, which ame-
5 liorates the underestimation problem of SD. This significant improvement in SD is due to the high spatial and temporal resolutions of observations.

The best combinations of snow parameters optimized for SK can be used to improve the snowfall prediction. Our re-
10 sults showed improvement in all snow variables in terms of RMSE by 3.3 %, 6.2 %, and 17.0 % for FSC, SA, and SD, respectively. Furthermore, SD increased after optimization, which led to increases in both soil temperature and sensible heat flux via an insulating response; soil moisture also in-
15 creased due to increased SD in previous years. This implies that the optimized snow parameters not only let the model solutions close to the observations, but also act in a physically consistent manner. Satellite observations proved to be effective in the optimization; however, their coarse resolution as
well as insufficient number of stations used for optimization often restrict improvement in the snow variables, as shown in some discouraging statistics including the mean bias and the coefficient of determination ($R^2$).

Based on the encouraging optimization results in the of-
25 fline Noah LSM, we plan to optimize the Noah LSM in a coupled land–atmosphere prediction system. The online Noah LSM can produce a spatial distribution of model variables over the land surface, which allows a two-dimensional assessment of model performance and a three-dimensional ex-
30 tension through various interactions between the land surface and the atmosphere. We anticipate that the optimized snow parameters can lead to positive effects on the atmospheric variables through the changes in heat fluxes as well as snow

variables in the Noah LSM. As a result, we can identify how optimal parameters are appreciated in SK in terms of both 35 horizontal and vertical distributions. Furthermore, the micro-GA–Noah LSM coupled system can be utilized to optimize other parameters in the Noah LSM, including the ones that indirectly affect the snow processes.

*Code availability.* The current version of the Noah LSM pro- 40 vided by National Center for Atmospheric Research (NCAR) is available from the website: https://ral.ucar.edu/solutions/products/ unified-wrf-noah-lsm (last access: 24 October 2022) (National Center for Atmosphere Research, 2022). The current version of the GA developed by David L. Carroll is available from the web- 45 site: https://cuaerospace.com/products-services/genetic-algorithm/ ga-drive-free-download (last access: 24 October 2022) (Carroll, 2022). The exact versions of the Noah LSM and GA used in this study are archived at https://doi.org/10.5281/zenodo.6873384 (Lim et al., 2021). They also contain the forcing data and output files of 50 the Noah LSM and micro-GA–Noah LSM coupled system and the scripts to plot the same figures as in this paper.

*Data availability.* The 1-hourly forcing data (i.e., ASOS) for the Noah LSM are obtained from the Open MET Data Portal, which is available at https://data.kma.go.kr/data/grnd/selectAsosRltmList. 55 do?pgmNo=36 (last access: 24 October 2022) (Korea Meteorological Administration, 2022), and ERA5-Land is provided by the Copernicus Climate Change Service (C3S) Climate Data Store (CDS), which is available at https://doi.org/10.24381/cds.e2161bac (Muñoz-Sabater, 2019). The snow depth is also obtained from 60 the Open MET Data Portal. The daily fractional snow cover and snow albedo from the MODIS/Terra Snow Cover Daily L3 Global 500 m SIN Grid, version 61, are available at https://doi.org/10.5067/ MODIS/MOD10A1.061 (Hall and Riggs, 2021).

*Author contributions.* SuL, SKP, HJG, WYL, YHL, and CC contributed to the conceptualization. SuL, SKP, and CC designed the experiments, and SuL carried them out with the investigation. SuL, HJG, and EL developed the model code, and EL and SeL contributed to the validation. SuL prepared the manuscript with contributions from all the co-authors.

*Competing interests.* The contact author has declared that none of the authors has any competing interests.

*Acknowledgements.* We are grateful to the managing editor and three anonymous referees for their valuable comments.

*Financial support.* This work is supported by the Basic Science Research Program through the National Research Foundation of Korea (NRF) funded by the Ministry of Education (grant no. 2018R1A6A1A08025520) and Development of Numerical Weather Prediction and Data Application Techniques (grant no. NTIS-1365003222) funded by the Korea Meteorological Administration. It is partly supported by an NRF grant funded by the Korean government (MSIT) (grant no. NRF-2021R1A2C1095535).

*Review statement.* This paper was edited by Yuefei Zeng and reviewed by three anonymous referees.

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

## Remarks from the typesetter

TS1 Please give an explanation of why this needs to be changed. We have to ask the handling editor for approval. Thanks.

TS2 Please see previous remark.

TS3 Please see previous remark.

TS4 Please see previous remark.

TS5 Please give an explanation of why this needs to be changed. We have to ask the handling editor for approval. Thanks.

TS6 Please see the previous remark.

TS7 Please see the previous remark.

TS8 Please see the previous remark.

TS9 Please see the previous remark.

TS10 Please see the previous remark.

TS11 Please see the previous remark.