# Peer review of "Optimization of Snow-Related Parameters in Noah Land Surface Model (v3.4.1) Using Micro-Genetic Algorithm (v1.7a)"

_Geoscientific Model Development, 2021_

## Referee Comment (RC1)

The manuscript "Optimization of Snow-Related Parameters in Noah Land Surface Model (v3.4.1) Using Micro-Genetic Algorithm (v1.7a)" by Lim et al. addresses an important problem of model tuning/optimization. However, the results are not very encouraging, it shows very small improvements. Moreover, the manuscript seriously lacks in its analysis/validation part. Authors should come up with more results/analysis to claim substantial improvements in their method. The following are the comments, which may improve the manuscript.

1) The improvements looks very small compare to the existing mean bias (table 4). The improvement ratio (equation 7), a metric used here gives an impression of big improvement, but in reality it is not so. For an example, improvement of RMSE from 6 to 5 will show about 16.5% improvements, but RMSE of 5 is still big. Statistically how significant are these improvements ? Pls put significance level.

2) I would be interested to see some more graphical representations of analysis, rather than many statistical number presented here. There are so many numbers/numerical values mentioned in the manuscript (particularly the results). It is very hard to recognise changes in the box plot (Figure 4), as the improvements are really minute.

3)  Pls write what is shown in the y-axis in Figure 4

4) I found the validation part of the manuscript is very weak. Perhaps you need to do more simulations/analysis to establish that your optimization method works better that the default model.

5) In several previous studies it has been shown that improvement/incorporation of real physical processes, such as discrete treatment of snow layer, more realistic snow physics significantly improves simulation of snow (e..g. Niu et al., 2011; Saha et al., 2017). Does your optimization fares better than above ?

6) Apart from RMSE, authors may also show any improvements in the correlation skill

7) How the seasonal cycle of snow parameters looks like (model vs observations)? Do you see improvements there also ?

8) What are the effects of optimized model on skin and sub-surface temperature, soil moisture, surface energy balance etc ?

9)  As mentioned in the beginning,  the ultimate goal is to improve forecast of snow over SK, I believe all-grid point simulation (gridded) would be a better strategy to really demonstrate the usefulness of this method.

Niu, G.-Y., et al. (2011), The community Noah land surface model with multiparameterization options (Noah-MP): 1. Model description and evaluation with local-scale measurements, *J. Geophys. Res.*, 116, D12109, doi:10.1029/2010JD015139.

Saha, S. K., K. Sujith, S. Pokhrel, H. S. Chaudhari, and A. Hazra (2017), Effects of multilayer snow scheme on the simulation of snow: Offline Noah and coupled with NCEP CFSv2, *Journal of Advances in Modeling Earth Systems*, 9, 271-290, doi:10.1002/2016MS000845.

---

## Author Response (AR1)

Response to the Referee 1 for the Manuscript gmd-2021-333
**"Optimization of Snow-Related Parameters in Noah Land Surface Model (v3.4.1) Using Micro-Genetic Algorithm (v1.7a)"**
by Sujeong Lim, Hyeon-Ju Gim, Ebony Lee, Seungyeon Lee, Won Young Lee, Yong Hee Lee, Claudio Cassardo, and Seon Ki Park

*The manuscript "Optimization of Snow-Related Parameters in Noah Land Surface Model (v3.4.1) Using Micro-Genetic Algorithm (v1.7a)" by Lim et al. addresses an important problem of model tuning/optimization. However, the results are not very encouraging, it shows very small improvements. Moreover, the manuscript seriously lacks in its analysis/validation part. Authors should come up with more results/analysis to claim substantial improvements in their method. The following are the comments, which may improve the manuscript.*

⇒ We appreciate the valuable and constructive comments, which helped us improve the quality of the manuscript. We have included more analysis/validation to enhance the results. Unfortunately, we found that there was a mistake when we simulated some stations (urban and built-up lands (UB) in OPT_5 and cropland (CL) in OPT_6), thus we corrected the statistical values in the manuscripts. An item-by-item response to the comments is provided below.

1. *The improvements looks very small compare to the existing mean bias (table 4). The improvement ratio (equation 7), a metric used here gives an impression of big improvement, but in reality it is not so. For an example, improvement of RMSE from 6 to 5 will show about 16.5% improvements, but RMSE of 5 is still big. Statistically how significant are these improvements? Pls put significance level.*

   ⇒ We agree the improvement ratio may emphasize itself, even for the small changes. Nevertheless, the improvement ratio helps to objectively determine how much change has occurred in the value. To recognize the original magnitude of them, we included the RMSE value of CNTL in the caption of Table R1 (Table 4 in the revised manuscript) below. In addition, the CNTL and OPTM (e.g., OPT_5 and OPT_6) experiments exhibit statistically significant linear relationships in the 95 % significance level. We have added this description in the caption of Table R1 (Table 4 in the revised manuscript) in the revised manuscript.

Table R1 (Table 4 in the revised manuscript): Improvement ratio (%) in RMSE, coefficient of determination ($R^2$), and mean bias (MB) of snow variables from CNTL to OPT_5, and OPT_6 over the ten representative stations. The statistic values in CNTL are following: RMSE is 0.270 for FSC, 0.155 for SA, and 10.599 for SD; $R^2$ is 0.219 for FSC, 0.183 for SA, and 0.806 for SD; MB is -0.107 for FSC, 0.0513 for SA and -5.38 cm for SD. The CNTL and OPTM (e.g., OPT_5 and OPT_6) experiments exhibit statistically significant linear relationships at the 95 % significance level.

| EXP | OPT_5 | | | OPT_6 | | |
|---|---|---|---|---|---|---|
| Snow Variable | FSC | SA | SD | FSC | SA | SD |
| RMSE | 1.3 % | 6.7 % | 13.8 % | 6.5 % | 8.5 % | 17.7 % |
| $R^2$ | 3.1 % | -2.4 % | 1.6 % | 16.4 % | -0.2 % | 3.0 % |
| MB | -31.8 % | 28.5 % | 40.9 % | -19.6 % | 32.6 % | 45.1 % |

2. *I would be interested to see some more graphical representations of analysis, rather than many statistical number presented here. There are so many numbers/numerical values mentioned in the manuscript (particularly the results). It is very hard to recognise changes in the box plot (Figure 4), as the improvements are really minute.*

⇒ We agree that the additional graphical representations are necessary to easily understand the changes between CNTL and OPTM experiments. Thus, we have included the scatter plots for the observation and simulation results with the RMSE and $R^2$ to help to understand Figure 4 (Figure R1 (Figure 5 in the revised manuscript) below). Since the observation patterns are different for different stations, we selected the representative station as for each land cover type: Ulleungdo (UL) for deciduous broadleaf forest (DBF), Gumi (GM) for mixed forest (MF), Bukgangneong (NG) for woody savanna (WS), Boryeong (BR) for cropland (CL), and Seoul (SL) for urban and built-up lands (UB). Firstly, the overall fractional snow cover (FSC) relatively are hard to recognize the explicit bias patterns in the scatter plots; however, GM in MF shows increasing FSC to solve the underestimated problems. Most statistics indicate the improved RMSE and $R^2$ from the CNTL to OPT_5 and additionally improved in OPT_6. Secondly, snow albedo (SA) is overestimated in CNTL and it is reduced in OPT_5 and OPT_6. For instance, UL in DBF shows decreasing SA in OPT_5 and following OPT_6. Lastly, snow depth (SD) is optimized using the hourly in-situ observations (i.e., more data), and hence shows remarkable improvement compared to FSC and SA, both using the daily satellite observations. Most stations have recovered the under-estimated SD with decreasing RMSE and increasing $R^2$. We include related descriptions in L330-345 (written in blue fonts) with Fig. R1 (Figure 5 in the revised manuscript).

"To understand more details of the improvements due to the optimization, we analyzed the scatter plots of observations versus model results along with the values of RMSD and $R^2$ (Figure 5). Since the observation patterns differ depending on their stations, we selected the representative station for each land cover type: Firstly, the overall FSC relatively is hard to recognize the explicit bias patterns in the scatter plots (Fig. 5(a), (d), (g), (j), and (m)); however, statistics indicate the improved RMSE from the CNTL to OPT_5 and additionally improved in OPT_6. As for the $R^2$, most stations show the largest value in OPT_6 except the NG for WS and BR for CL. In particular, GM in MF shows increasing FSC in OPT_6 to solve the underestimated problems with the best RMSE and $R^2$. Secondly, SA is overestimated in CNTL, and it is reduced in OPT_5 and OPT_6. For instance, UL in DBF shows decreasing SA in OPT_5 and following OPT_6 (Fig. 5(b)); it also shows the best RMSE and $R^2$ performance. Most stations show the smallest RMSE in OPT_6 and a larger $R^2$ in OPT_5 or OPT_6 (Fig. 5(b), (e), (h) and (k)); however, SL in UB was deteriorated RMSE and $R^2$ after optimization (Fig. 5(n)). Lastly, SD is optimized using the hourly in-situ observations (i.e., more data) and hence shows remarkable improvement compared to FSC and SA, both using the daily satellite observations. For example, UL in DBF results in a notable increase in the underestimated SD with the lowest RMSE and the highest $R^2$ (Fig. 5(c)). It is hard to say which optimization experiment has the best results, but the optimization performance is usually better than CNTL in terms of RMSD (e.g., UL for DBF, GM for MF, NG for WS, SL for UB) and $R^2$ (e.g., UL for DBF, GM for MF, and BR for CL). As a result, most stations in OPT_5 and OPT_6 are generally closer to observations than CNTL, and OPT_6 leads the lowest RMSE and the highest $R^2$ in all snow-related variables."

3. *Pls write what is shown in the y-axis in Figure 4*

   $\Rightarrow$ We added the y-axis information (Fig. R2 (Figure 4 in the revised manuscript)) as follows: (a) FSC bias, (b) SA bias, and (c) SD bias (cm). The wrong maximum and mean value of each bias in OPT_5 and OPT_6 have been corrected in the caption.

4. *I found the validation part of the manuscript is very weak. Perhaps you need to do more simulations/analysis to establish that your optimization method works better that the default model.*

   $\Rightarrow$ We prepared additional analyses with the scatter plots for snow variables (Fig. R1 (Figure 5 in the revised manuscript)), as mentioned in #2 above, and the time series of secondary variables (e.g., soil temperature, soil moisture, and sensible heat flux) through the snow optimization (Fig. R3 (Figure 6 in the revised manuscript) with L346-354 (blue fonts below)).

   "Lastly, we have investigated how the optimized snow parameters can effect on the other variables in LSM. Figure 6 is the time series of the

differences of LSM variables (e.g., soil temperature, sensible heat flux, and soil moisture) between OPT_6 and CNTL (i.e., OPT_6 minus CNTL) following SD changes. Although they are not directly optimized, they respond to the optimized snow parameters through associated physical processes. For example, soil temperature in the first soil layer (7 cm) increases as SD increases after optimization, which consequently increases sensible heat flux. The residual of surface energy balance is close to zero (not shown), implying that the surface energy balance is well conserved even after optimization. Soil moisture depends on snow melt, following the trend of increased snowfall in the previous winter. Extreme fluctuations sometimes appear in the time series analyses due to nonlinear effects, but we can understand the overall tendency according to the increased SD in the land surface."

As the off-line Noah LSM is one-dimensional, it requires lots of computing time for simulations and verifications at all the grid points. We plan to address more stations in our further study. Moreover, we also plan to optimize the Noah LSM in a coupled land-atmosphere prediction system to produce two-dimensional data in one model run. These explanations have added in the revised manuscript (L370-371; L379-384).

"As the further study, the online Noah LSM can help to include more observation stations by covering the all grid points over SK."

"Based on the encouraging optimization results in the off-line Noah LSM, we plan to optimize the Noah LSM in a coupled land-atmosphere prediction system. The online Noah LSM can produce a spatial distribution of model variables over the land surface, which allows a two-dimensional assessment of model performance. We anticipate the optimized snow parameters can lead to positive effects on the atmospheric variables through the changes of heat fluxes as well as snow variables in Noah LSM. As a result, we can identify how optimal parameters are appreciated in SK in terms of both horizontal and vertical distributions. In addition, our coupled system of micro-GA and Noah LSM can be utilized to optimize other parameters in Nosh LSM."

5. *In several previous studies it has been shown that improvement or incorporation of real physical processes, such as discrete treatment of snow layer, more realistic snow physics significantly improves simulation of snow (e.g., Niu et al., 2011; Saha et al., 2017). Does your optimization fares better than above?*

⇒ We agree with the reviewer that some previous studies have improved snow simulation through more realistic physical parameterization [1] or discrete treatment of snow layer [2]. We can develop more realistic parameterization schemes and make improvement in the model performance; however, those scheme are still under uncertainty, especially in parameter values. Moreover, the model performance by more realistic parameterization scheme may significantly improve in one region but it may less significantly improve or even deteriorate in other places, due to uncertainties in parameter values. *Parameter estimation* is not competing with the development of more realistic physical parameterization; it is rather an effort to further improve the model performance by reducing uncertainty in pre-existing parameterization schemes by optimizing the parameter values inside the schemes based on the observational data that reflect local characteristics. If the employed parameterization scheme has less uncertainty, improvement by parameter estimation on that scheme may not be significant; if the scheme has large uncertainty in parameter values, parameter estimation may bring about prominent improvement in the scheme's performance. Therefore, we believe that development of more realistic physical parameterization scheme, followed by appropriate parameter estimation, will create a strong synergy between them that results in higher model performance, as indicated in [3]. We have added these explanations in the revised manuscript (L359-364).

"This parameter estimation is an effort to further improve the model performance by reducing uncertainty in pre-existing parameterization schemes by optimizing the parameter values inside the schemes based on the observational data that reflect local characteristics to improve snow simulation. If the employed parameterization scheme has less uncertainty, improvement by parameter estimation on that scheme may not be significant; if the scheme has large uncertainty in parameter values, parameter estimation may bring about prominent improvement in the scheme's performance."

6. *Apart from RMSE, authors may also show any improvements in the correlation skill*

   $\Rightarrow$ We included the coefficient of determination ($R^2$), which measures the proportion of variation for a dependent variable that can be explained by an independent variable, in Table R1 (Table 4 in the revised manuscript). Like the RMSE, the $R^2$ of FSC and SD also improved in OPTM. The SA was weakly worsened in OPT_5, but it was almost recovered to the CNTL in OPT_6. The related explanations have contained in the revised manuscript (L294-296; L311-312; L328-329).

   "The performance has been evaluated using the improvement ratio, which indicates how much the RMSE, MB, and coefficient of determination ($R^2$) of optimized experiments (i.e., OPT_5, OPT_W, and OPT_6) is improved compared to CNTL, as shown in Eq. (7) (Table 4)."

   "We also investigated the $R^2$, which measures the proportion of variation for a dependent variable that can be explained by an independent variable. As a result, the OPT_5 improves the 3.1 % and 1.6 % for FSC and SD while deteriorates 2.4 % for SA."

   "Like the RMSE, the $R^2$ of FSC and SD also improved in OPT_5 and OPT_6. The SA worsened in OPT_5 was almost recovered to the CNTL in OPT_6."

7. *How the seasonal cycle of snow parameters looks like (model vs observations)? Do you see improvements there also ?*

   ⇒ Snow parameters do not have the observations; thus, it is impossible to compare the snow-related parameters between model and observations. In addition, the snow is found over South Korea only in the wintertime, so it is hard to identify the seasonable cycle of snow parameters in our study.

8. *What are the effects of optimized model on skin and sub-surface temperature, soil moisture, surface energy balance etc?*

   ⇒ We investigate the responses of secondary variables due to optimization of snow parameter (Fig. R3 (Figure 6 in the revised manuscript)). We bring the results of UL in DBF which shows enhancements on all of snow variables in Fig. R1 (Figure 5 in the revised manuscript). Increased SD warms the soil temperature in the first soil layer (7 cm) through the land surface insulative response, resulting in larger sensible heat flux. The residual of the surface energy balance equation gets close zero, thus the surface energy balance is conserved after optimization (Figure is not shown). Finally, the soil moisture depends on the snow melt, hence it follows the increased snowfall in the previous winter. Because this is an hourly data, extreme fluctuations sometimes appear in the time series analyses, but we can understand the overall tendency from the increased SD. The related descriptions are added in the revised manuscript (L346-354, blue fonts below).

   "Lastly, we have investigated how the optimized snow parameters can effect on the other variables in LSM. Figure 6 is the time series of the differences of LSM variables (e.g., soil temperature, sensible heat flux, and soil moisture) between OPT_6 and CNTL (i.e., OPT_6 minus CNTL) following SD changes. Although they are not directly optimized, they respond to the optimized snow parameters through associated physical processes. For example, soil temperature in the first soil layer (7 cm) increases as SD increases after optimization, which consequently increases sensible heat flux. The residual of surface energy balance is close to zero (not shown), implying that the surface energy balance is well conserved even after optimization. Soil moisture depends on snow melt, following the trend of increased snowfall in the previous winter. Extreme fluctuations sometimes appear in the time series analyses due to nonlinear effects, but we can understand the overall tendency according to the increased SD in the land surface."

9. *As mentioned in the beginning, the ultimate goal is to improve forecast of snow over SK, I believe all-grid point simulation (gridded) would be a better strategy to really demonstrate the usefulness of this method.*

   ⇒ We fully agree with the reviewer. As mentioned in #4 above, running the off-line Noah LSM over all grid points requires a large amount of

computational time. Thus, we have sampled representative stations in this study for effective optimization. Following the reviewer's suggestion, we will do simulations over all the grid points in our further study. Based on the promising results using the off-line Noah LSM, we have a plan to optimize the Noah LSM in a coupled land-atmosphere prediction system (e.g., Weather Research and Forecasting (WRF)-Noah LSM). While the off-line Noah LSM is a one-dimensional column model, the Noah LSM coupled to WRF is able to simulate the two-dimensional features with prescribed spatial resolution. Moreover, it can interact with not only the multiple soil layers but also the atmospheric layers. As a further study, we anticipate the optimized snow parameters can lead to forecast improvement in the atmospheric variables through the changes of heat fluxes as well as snow variables in the LSM. These explanations have included in the revised manuscript (L379-384).

"Based on the encouraging optimization results in the off-line Noah LSM, we plan to optimize the Noah LSM in a coupled land-atmosphere prediction system. The online Noah LSM can produce a spatial distribution of model variables over the land surface, which allows a two-dimensional assessment of model performance. We anticipate the optimized snow parameters can lead to positive effects on the atmospheric variables through the changes of heat fluxes as well as snow variables in Noah LSM. As a result, we can identify how optimal parameters are appreciated in SK in terms of both horizontal and vertical distributions. In addition, our coupled system of micro-GA and Noah LSM can be utilized to optimize other parameters in Nosh LSM."

**References**

[1] Niu, G.-Y., et al.: The community Noah land surface model with multiparameterization options (Noah-MP): 1. Model description and evaluation with local-scale measurements, J. Geophys. Res., 116, D12109, doi:10.1029/2010JD015139, 2011.

[2] Saha, S. K., Sujith K., Pokhrel S., Chaudhari H. S., and Hazra A.: Effects of multilayer snow scheme on the simulation of snow: Offline Noah and coupled with NCEP CFSv2, J. Adv. Model. Earth Sy., 9, 271-290, doi:10.1002/2016MS000845, 2017.

[3] Duan, Q., Di, Z., Quan, J., Wang, C., Gong, W., Gan, Y., Ye, A., Miao, C., Miao, S., Liang, X., and Fan, S.: Automatic Model Calibration: A New Way to Improve Numerical Weather Forecasting, Bull. Am. Meteorol. Soc., 98, 959-970, 2017.

[Figure]

Figure R1 (Figure 5 in the revised manuscript): Scatter plots for the observation (OBS) and land surface model (LSM) results: CNTL (red), OPT_5 (blue) and OPT_6 (green). The representative station in each land cover type are analyzed such as (a)-(c) DBF: UL, (d)-(f) MF: GM, (g)-(i) WS: NG, (j)-(l) CL: BR, (m)-(o) UB: SL. From the left to right panels, they are the FSC, SA, and SD (cm). Compared to observations, the statistics (e.g., RMSE and $R^2$) in each experiment are indicated in each panel.

[Figure]

Figure R2 (Figure 4 in the revised manuscript): Box plots of (a) FSC bias, (b) SA bias, and (c) SD bias (cm) for CNTL, OPT_5 and OPT_6. The maximum differences are indicated with the black star symbol (e.g., 0.637 (CNTL), 0.643 (OPT_5), 0.570 (OPT_6) for FSC, 0.605 (CNTL), 0.563 (OPT_5), and 0.525 (OPT_6) for SA, and 34.1 cm (CNTL), 45.1 cm (OPT_5), and 46.3 cm (OPT_6) for SD). Each mean of snow variables is indicated as a black circle (e.g., -0.107 (CNTL), -0.125 (OPT_5), and -0.130 (OPT_6) for FSC, 0.0513 (CNTL), 0.0381 (OPT_5), and 0.0359 (OPT_6) for SA, and -5.38 cm (CNTL), -3.46 cm (OPT_5), and -2.93 cm (OPT_6) for SD).

[Figure]

Figure R3 (Figure 6 in the revised manuscript): Time series of difference between CNTL to OPT_6 for the UL in DBF during the May 2009 to April 2018: (a) SD (cm), (b) soil temperature at the top soil layer (ST; 7 cm) (K), (c) sensible heat flux (SH; W m$^{-2}$), (d) soil moisture at the top soil layer (7 cm) (SM; m$^3$ m$^{-3}$).

Response to the Referee 2 for the Manuscript gmd-2021-333
**"Optimization of Snow-Related Parameters in Noah Land Surface Model (v3.4.1) Using Micro-Genetic Algorithm (v1.7a)"**
by Sujeong Lim, Hyeon-Ju Gim, Ebony Lee, Seungyeon Lee, Won Young Lee, Yong Hee Lee, Claudio Cassardo, and Seon Ki Park

*This study worked on determining the optimal parameter values in the snow-related processes – snow cover fraction, snow albedo, and snow depth – of the Noah LSM, using the micro-genetic algorithm and the in-situ surface observations and remotely-sensed satellite data. The study area was South Korea. This manuscript does not have sufficient elements on the model development, it is rather a study of applying a certain optimization algorithm to calibrate the model parameters. I have doubts about the novelty of this manuscript and its suitability for consideration for publication in Geoscientific Model Development. Below are some comments which I hope could help improve the manuscript.*

⇒ We really appreciate the valuable and constructive comments, which helped us improve the quality of the manuscript. Our study is a parameter estimation problem, which is based on the assumption that all the physical parameterization schemes are not perfect and have uncertainties, especially in their parameter values; thus, it is strongly and directly linked to 'assessment of model performance' through parameterization schemes, which corresponds to the scope of Geoscientific Model Development (GMD). Parameter estimation is a companion of parameterizations as it reduces the uncertainties in the parameter values of newly-developed parameterization schemes and enhances the model performance through the schemes; furthermore, a new method of comparing model results with observational data is developed in our study through various fitness functions in the course of optimization. In this sense, we believe that our study also indirectly satisfies the scopes of GMD, described as 'developments such as new parameterizations' as well as 'developing novel ways of comparing model results with observational data'. We have faithfully followed the reviewer's suggestions and included more analysis/validation to enhance the results. An item-by-item response to the comments is provided below.

1. *Short Introduction and unclear novelty of this study. The introduction is rather short and the novelty of this study is not explicitly stated.*

   ⇒ We appreciate the reviewer pointing this out. We have revised a paragraph in *Introduction* (L35-53 in the revised manuscript) by adding more statements as follows (see blue parts):

   "Uncertainties in parameterized physical processes have been observed and quantified in various numerical models (e.g., Mallet and Sportisse, 2006; Gubler et al., 2012; Shutts and Pallarès, 2014; Folberth et al., 2019; Li et al., 2020; Olafsson and Bao, 2020; Pathak et al., 2020; Souza et al.,

2020). Such uncertainties can be also reduced by estimating optimal parameter values in the subgrid-scale parameterization schemes (e.g., Annan and Hargreaves, 2004; Lee et al., 2006; Neelin et al., 2010; Yu et al., 2013; Zhang et al., 2015; Kotsuki et al., 2018; Liet al., 2018; Chinta and Balaji, 2020). Because empirical parameters are commonly derived from the observations or theoretical calculations, their estimated values are strongly dependent on the local characteristics of the region and period where the observations are made. Thus, *parameter estimation* that fits the model outputs to the observations is essentially required to obtain an adequate parameter [1]. It may be done using a *trial and error* approach by manual, but the *optimization algorithm* helps to replace enormous experiments by automatically minimizing the difference between model and observations [2]. For example, a global optimization tool, called the micro-genetic algorithm (micro-GA), has been effectively used for estimating the optimal parameter values (e.g., Yu et al., 2013) and for finding the optimal set of parameterization schemes (e.g., Hong et al., 2014, 2015; Park and Park, 2021).

Most snow processes in the LSMs are parameterized based on the observations in specific local regions, and hence they may not represent adequately the situation in SK and be the source of uncertainties for numerical snow prediction over SK. We aim at obtaining the optimal parameter values of the snow-related processes — snow cover, snow albedo, and snow depth — in a LSM using the micro-GA, which causes better LSM performance over SK. This study represents the first attempt to develop a coupled system of micro-GA and Noah LSM for parameter estimation of the snow processes. Section 2 describes the methodology, including the snow processes of the LSM and the micro-GA optimization tool. Section 3 explains experiment design. Results, discussion and conclusions are provided in sections 4, 5 and 6, respectively."

2. *Insufficient details on the methods/procedures. Section 2.2 and Table 2 miss necessary details on the selected parameters and settings for the different experiments.*

⇒ In Section 2.2, we have focused on the GA algorithm itself and the fitness function. Descriptions on the selected parameters and settings for different experiments are separately provided in Section 3. We modified Table 2 with the typo and list order correction to help understanding (Table R1 (Table 2 in the revised manuscript)). Moreover, we have added more details on the parameter settings in Section 3 (L238-246 in the revised manuscript) as follows (see blue parts):

Table R1 (Table 2 in the revised manuscript): The input parameters for micro-GA in experiments OPT_5 and OPT_W.

| Input Parameter | OPT_5 | OPT_W |
|---|---|---|
| Population size | 5 | 5 |
| Crossover operator | 1.0 | 1.0 |
| Elitism | on | on |
| Number of parameters | 5 | 1 |
| Number of chromosomes | 30 | 5 |
| Maximum value of generations | 200 | 100 |

"Table 2 describes the input parameters used in this study. We follow the options known as the best performance in micro-GA; it is done with a population size of 5 and a uniform crossover (i.e., crossover operator = 1.0) with elitism [3, 4, 5]. The uniform crossover makes all populations perform a crossover at every generation to acquire the diversity [6]. The number of parameters to be optimized is 5 for OPT_5 and 1 for OPT_W. The number of chromosomes determines the number of cases expressed in a binary format. For example, the selected parameters — $P_s$, $\alpha_{max,CofE}$, $C$, $P_1$, $P_2$, and $W_{max}$ — use different chromosomes, i.e., 5, 5, 5, 6, 4, and 5, respectively; thus, the total number of chromosomes is 30 for OPT_5 and 5 for OPT_6. The maximum value of generations at the end of optimization is generally set to 100 [4, 5, 7], whereas we increased generations up to 200 in OPT_5 due to larger number of parameters to be optimized."

3. *I advise the authors to add more figures to show the comparison, via scatter plot, time series plot to show the modelling results in different perspectives. Besides the RMSE value, what about the performance of the model in terms of other commonly used metrics such as R or $R^2$ value?*

$\Rightarrow$ Following the reviewer's comments, we have conducted additional analyses and added more figures, including scatter plots, times series, and statistics including $R^2$. Figure R1 (Figure 5 in the revised manuscript) (see below) represents the scatter plots of observations versus model results along with the values of RMSD and $R^2$. Consistent with the statistical results in the original manuscript, OPT_6 shows improved snow variables in the scatter plots for Ulleungdo (UL) in the deciduous broadleaf forest (DBF). In particular, compared to CNTL, optimization results in notable increase in the underestimated snow depth (SD; Fig R1 (Figure 5 in the revised manuscript) (c)) and negligible changes in fractional snow cover (FSC; Fig R1 (Figure 5 in the revised manuscript) (a)) and snow albedo (SA; Fig R1 (Figure 5 in the revised manuscript) (b)). In statistical analyses, represented by RMSE and $R^2$, OPT_5 and OPT_6 are generally closer to observations than CNTL while OPT_6 shows the lowest RMSE and the highest $R^2$. We have added the scatter diagrams and statistical analyses for other stations and land cover types in the revised manuscript (see Fig. R1 (Figure 5 in the revised manuscript) therein with L330-345 written in blue fonts).

"To understand more details of the improvements due to the optimization, we analyzed the scatter plots of observations versus model results along with the values of RMSD and $R^2$ (Figure 5). Since the observation patterns differ depending on their stations, we selected the representative station for each land cover type: Firstly, the overall FSC relatively is hard to recognize the explicit bias patterns in the scatter plots (Fig. 5(a), (d), (g), (j), and (m)); however, statistics indicate the improved RMSE from the CNTL to OPT_5 and additionally improved in OPT_6. As for the $R^2$, most stations show the largest value in OPT_6 except the NG for WS and BR for CL. In particular, GM in MF shows increasing FSC in OPT_6 to solve the underestimated problems with the best RMSE and $R^2$. Secondly, SA is overestimated in CNTL, and it is reduced in OPT_5 and OPT_6. For instance, UL in DBF shows decreasing SA in OPT_5 and following OPT_6 (Fig. 5(b)); it also shows the best RMSE and $R^2$ performance. Most stations show the smallest RMSE in OPT_6 and a larger $R^2$ in OPT_5 or OPT_6 (Fig. 5(b), (e), (h) and (k)); however, SL in UB was deteriorated RMSE and $R^2$ after optimization (Fig. 5(n)). Lastly, SD is optimized using the hourly in-situ observations (i.e., more data) and hence shows remarkable improvement compared to FSC and SA, both using the daily satellite observations. For example, UL in DBF results in a notable increase in the underestimated SD with the lowest RMSE and the highest $R^2$ (Fig. 5(c)). It is hard to say which optimization experiment has the best results, but the optimization performance is usually better than CNTL in terms of RMSD (e.g., UL for DBF, GM for MF, NG for WS, SL for UB) and $R^2$ (e.g., UL for DBF, GM for MF, and BR for CL). As a result, most stations in OPT_5 and OPT_6 are generally closer to observations than CNTL, and OPT_6 leads the lowest RMSE and the highest $R^2$ in all snow-related variables."

In Fig R2 (Figure 6 in the revised manuscript), we analyzed the time series of the differences of secondary variables (e.g., soil temperature, soil moisture, and sensible heat flux) between OPT_6 and CNTL (i.e., OPT_6 minus CNTL). Although these variables are not directly optimized, they respond to the optimized snow parameters through associated physical processes. For example, soil temperature in the first soil layer (7 cm) increases as SD increases after optimization, which consequently increases sensible heat flux. The residual of surface energy balance is close to zero (not shown), implying that the surface energy balance is well conserved even after optimization. Soil moisture depends on snow melt, following the trend of increased snowfall in the previous winter. Extreme fluctuations sometimes appear in the time series analyses due to nonlinear effects, but we can understand the overall tendency according to the increased SD

in the land surface. The related descriptions are added in the revised manuscript (L346-354, blue fonts below).

"Lastly, we have investigated how the optimized snow parameters can effect on the other variables in LSM. Figure 6 is the time series of the differences of LSM variables (e.g., soil temperature, sensible heat flux, and soil moisture) between OPT_6 and CNTL (i.e., OPT_6 minus CNTL) following SD changes. Although they are not directly optimized, they respond to the optimized snow parameters through associated physical processes. For example, soil temperature in the first soil layer (7 cm) increases as SD increases after optimization, which consequently increases sensible heat flux. The residual of surface energy balance is close to zero (not shown), implying that the surface energy balance is well conserved even after optimization. Soil moisture depends on snow melt, following the trend of increased snowfall in the previous winter. Extreme fluctuations sometimes appear in the time series analyses due to nonlinear effects, but we can understand the overall tendency according to the increased SD in the land surface."

We also added $R^2$ in Table R2 (Table 4 in the revised manuscript) below. Both FSC and SD showed improvement in terms of RMSE and $R^2$. The SA worsened in OPT_5 but it showed less deterioration in OPT_6, getting closer to CNTL in terms of $R^2$. The related explanations have contained in the revised manuscript (L294-296; L311-312; L328-329).

"The performance has been evaluated using the improvement ratio, which indicates how much the RMSE, MB, and coefficient of determination ($R^2$) of optimized experiments (i.e., OPT_5, OPT_W, and OPT_6) is improved compared to CNTL, as shown in Eq. (7) (Table 4)."

"We also investigated the $R^2$, which measures the proportion of variation for a dependent variable that can be explained by an independent variable. As a result, the OPT_5 improves the 3.1 % and 1.6 % for FSC and SD while deteriorates 2.4 % for SA."

"Like the RMSE, the $R^2$ of FSC and SD also improved in OPT_5 and OPT_6. The SA worsened in OPT_5 was almost recovered to the CNTL in OPT_6."

Table R2 (Table 4 in the revised manuscript): Improvement ratio (%) in RMSE, coefficient of determination ($R^2$), and mean bias (MB) of snow variables from CNTL to OPT_5, and OPT_6 over the ten representative stations. The statistic values in CNTL are following: RMSE is 0.270 for FSC, 0.155 for SA, and 10.599 for SD; $R^2$ is 0.219 for FSC, 0.183 for SA, and 0.806 for SD; MB is -0.107 for FSC, 0.0513 for SA and -5.38 cm for SD. The CNTL and OPTM (e.g., OPT_5 and OPT_6) experiments exhibit statistically significant linear relationships at the 95 % significance level.

| EXP | OPT_5 | | | OPT_6 | | |
|---|---|---|---|---|---|---|
| Snow Variable | FSC | SA | SD | FSC | SA | SD |
| RMSE | 1.3 % | 6.7 % | 13.8 % | 6.5 % | 8.5 % | 17.7 % |
| $R^2$ | 3.1 % | -2.4 % | 1.6 % | 16.4 % | -0.2 % | 3.0 % |
| MB | -31.8 % | 28.5 % | 40.9 % | -19.6 % | 32.6 % | 45.1 % |

4. *Results need more description and particularly figures. I advise adding more figures on the modelling results, and particularly representing the spatial patterns of modeling results. The author studied South Korea, readers are interested in the spatial distribution of model performance.*

⇒ We appreciate the reviewer's valuable comment. As the Noah LSM is a one-dimensional column model, we should run the off-line Noah LSM over all the grid point by point, which requires a large amount of computational time. Thus, we have sampled representative stations in this study for effective optimization. Based on the promising optimization results in the off-line Noah LSM, we plan to extend our study to optimize the online mode of Noah LSM, coupled to an atmospheric model (e.g., WRF). Then, we will be able to assess the model performance in terms of spatial distributions, and we will do more experiments following the reviewer's comment in our follow-up study. The related descriptions are added in the revised manuscript (L370-371; L379-384, blue fonts below).

"As the further study, the online Noah LSM can help to include more observation stations by covering the all grid points over SK."

"Based on the encouraging optimization results in the off-line Noah LSM, we plan to optimize the Noah LSM in a coupled land-atmosphere prediction system. The online Noah LSM can produce a spatial distribution of model variables over the land surface, which allows a two-dimensional assessment of model performance. We anticipate the optimized snow parameters can lead to positive effects on the atmospheric variables through the changes of heat fluxes as well as snow variables in Noah LSM. As a result, we can identify how optimal parameters are appreciated in SK in terms of both horizontal and vertical distributions. In addition, our coupled system of micro-GA and Noah LSM can be utilized to optimize other

parameters in Nosh LSM."

5. *Discussion is completely missing. The current manuscript has no discussion. I strongly advise the authors to compare their findings with existing literature. In addition, what are the limitation of the study? And any potential solutions for future studies? What are effects of some settings or input on the modelling results? Lots of aspects need to be discussed.*

⇒ We have included the *Discussion* section before the *Conclusion* as follows (L355-384):

"Generally, the Noah LSM tends to simulate less snow amount during the peak winter and earlier snow melting, and consequently overestimates SA [8]. Our experiment with no optimization (CNTL) reveals underestimation of SD and FSC and overestimation of SA compared to the in-situ or satellite observations. We developed a coupled system of micro-GA and Noah LSM to reduce the uncertainties in parameterized snow processes through optimization of parameter values. This parameter estimation is an effort to further improve the model performance by reducing uncertainty in pre-existing parameterization schemes by optimizing the parameter values inside the schemes based on the observational data that reflect local characteristics to improve snow simulation. If the employed parameterization scheme has less uncertainty, improvement by parameter estimation on that scheme may not be significant; if the scheme has large uncertainty in parameter values, parameter estimation may bring about prominent improvement in the scheme's performance. Our results showed improvement in all snow variables in terms of RMSE by 6.5 %, 8.5 %, and 17.7 % for FSC, SA, and SD, respectively. Furthermore, SD increased after optimization, which lead to increases in both soil temperature and sensible heat flux due to insulating response; soil moisture also increased due to increased SD in previous years. This implies that the optimized snow parameters not only let the model solutions close to the observations but also act in a physically consistent manner. In case of some worsen statistics such as MB or $R^2$ in OPT_6, the insufficient stations used for optimization or a coarse resolution in satellite observation can limit to improve the snow variables. As the further study, the online Noah LSM can help to include more observation stations by covering the all grid points over SK. Moreover, we can optimize other parameters that indirectly affects to snow processes not only direct parameters used in this study.

The coupling system of micro-GA and Noah LSM automatically estimates the optimal snow-related parameters by objectively comparing observations and model solutions through the fitness function. Instead of trial-and-error procedures, it has an advantage to reduce a substantial amount of computational time. The original micro-GA reduces the computational

time using the elitism and re-initialization methods in the small number of individuals. We have developed a parallel system on the coupled system to further improve the computational efficiency in this study; it enables us to simultaneously execute multiple individuals in one generation and multiple Noah LSM runs in one individual.

Based on the encouraging optimization results in the off-line Noah LSM, we plan to optimize the Noah LSM in a coupled land-atmosphere prediction system. The online Noah LSM can produce a spatial distribution of model variables over the land surface, which allows a two-dimensional assessment of model performance. We anticipate the optimized snow parameters can lead to positive effects on the atmospheric variables through the changes of heat fluxes as well as snow variables in Noah LSM. As a result, we can identify how optimal parameters are appreciated in SK in terms of both horizontal and vertical distributions. In addition, our coupled system of micro-GA and Noah LSM can be utilized to optimize other parameters in Nosh LSM."

**References**

[1] Duan, Q., Di, Z., Quan, J., Wang, C., Gong, W., Gan, Y., Ye, A., Miao, C., Miao, S., Liang, X., and Fan, S.: Automatic Model Calibration: A New Way to Improve Numerical Weather Forecasting, Bull. Am. Meteorol. Soc., 98, 959-970, 2017.

[2] Duan Q., Schaake J., Andréassian V., Franks S., Goteti G., Gupta H. V., Gusev Y. M., Habets F., Hall A., Hay L., Hogue T., Huang M., Leavesley G., Liang X., Nasonova O. N., Noilhan J., Oudin L. , Sorooshian S., Wagener T., and Wood E. F.: Model Parameter Estimation Experiment (MOPEX): An overview of science strategy and major results from the second and third workshops, J. Hydrol., 320, 3-17, 2006.

[3] Carroll, D. L.: Genetic algorithms and optimizing chemical oxygen-iodine lasers, Developments in theoretical and applied mechanics, 18, 411–424, 1996.

[4] Yu, X., Park, S.K., Lee, Y.H., and Choi, Y.-S.: Quantitative precipitation forecast of a tropical cyclone through optimal parameter estimation in a convective parameterization. SOLA, 9, 36–39, 2013.

[5] Yoon J. W., Lim S., and Park S. K.: Combinational Optimization of the WRF Physical Parameterization Schemes to Improve Numerical Sea Breeze Prediction Using Micro-Genetic Algorithm. Appl. Sci., 11, 11221, https://doi.org/10.3390/app112311221, 2021.

[6] Lee, J., Kim, S. M., Park, H. S., and Woo, B. H.: Optimum design of cold-formed steel channel beams using micro Genetic Algorithm. Eng. Struct., 27, 17-24, 2005.

[7] Zhu, J., Shu, J., and Yu, X.: Improvement of typhoon rainfall prediction based on optimization of the Kain-Fritsch convection parameterization scheme using a micro-genetic algorithm. Front. Earth Sci, 13, 721–732, 2019.

[8] Saha, S. K., Sujith K., Pokhrel S., Chaudhari H. S., and Hazra A.: Effects of multilayer snow scheme on the simulation of snow: Offline Noah and coupled with NCEP CFSv2, J. Adv. Model. Earth Sy., 9, 271-290, doi:10.1002/2016MS000845, 2017.

[Figure]

Figure R1 (Figure 5 in the revised manuscript): Scatter plots for the observation (OBS) and land surface model (LSM) results: CNTL (red), OPT_5 (blue) and OPT_6 (green). The representative station in each land cover type are analyzed such as (a)-(c) DBF: UL, (d)-(f) MF: GM, (g)-(i) WS: NG, (j)-(l) CL: BR, (m)-(o) UB: SL. From the left to right panels, they are the FSC, SA, and SD (cm). Compared to observations, the statistics (e.g., RMSE and $R^2$) in each experiment are indicated in each panel.

[Figure]

Figure R2 (Figure 6 in the revised manuscript): Time series of difference between CNTL to OPT_6 for the UL in DBF during the May 2009 to April 2018: (a) SD (cm), (b) soil temperature at the top soil layer (7 cm) (ST; K), (c) Sensible heat flux (SH; W m$^{-2}$), (d) soil moisture at the top soil layer (7 cm) (SM; m$^3$ m$^{-3}$).

---

## Author Response (AR3)

Response to Comments by Topical Editor Decision: Publish Subject to Minor Revisions (Review by Editor) for the Manuscript gmd-2021-333 **"Optimization of Snow-Related Parameters in Noah Land Surface Model (v3.4.1) Using Micro-Genetic Algorithm (v1.7a)"** by Sujeong Lim, Hyeon-Ju Gim, Ebony Lee, Seungyeon Lee, Won Young Lee, Yong Hee Lee, Claudio Cassardo, and Seon Ki Park

*We appreciate the valuable and constructive comments, which helped us improve the quality of the manuscript. An item-by-item response to the comments is provided below.*

1. *Line 39-41: As a common knowledge, it is unnecessary to have this sentence "Here, the parameter is constant that makes up the equations, which is usually fixed during the simulation and differs from the variable representing the time-varying state of the model". Instead, I think that authors should make it clear in the text if the estimated parameters vary with space and time in this work. If yes, it would be nice to provide field plots of variables compared to observations (Currently, only plots with numbers are shown).*

    ⇒ Thank you for your comments. In this study, 'parameters' (i.e., $P_s$, $\alpha_{max,CofE}$, $C$, $P_1$, $P_2$, and $W_{max}$) are all constants while 'variables' (i.e., snow cover, snow albedo, and snow depth) vary with time and space. Therefore, we do not have any field plot showing temporal/spatial variations of parameters. Instead, we have added a statement in the revised manuscript, saying "These parameters are all constants and do not vary with time and space." (Line 231-232), and removed the statement in Line 39-41 as the Editor suggested.

2. *It should be said at the beginning of Line 83-86 that results are presented in Fig.1.*

    ⇒ Thank you for your comments. We have introduce the Fig. 1 in the revised manuscript (Line 86) at the beginning of original manuscript 83-86:

    "Figure 1 represents the responses of the snow variables to the variations in the snow-related parameters for given ranges."

3. *It would be good if authors can add a discussion on the current problems of old parameter estimation method at the end of 2.1 or at the beginning of 2.2.*

    ⇒ Thank you for your suggestion. We have included the old parameter estimation method at the end of 2.1 in the revised manuscript (Line 74-78).

    "The above-mentioned snow processes contain certain estimated coefficients or constants, known as *parameters*, which employ typical, empirical or a priori values. The parameters are provided as look-up tables based on their samples in the field or lab. Traditionally, they are tuned by trial

and error to calibrate the model against historical observations in a specific location; however, a systematic and objective procedure is essentially required for a large number of stations (Duan et al., 2006; Rosolem et al., 2013). We explain below the details of the snow-related parameters to be optimized for various stations in SK."

4. *Would be better to move the last paragraph of section 5 to the section of conclusion as outlook? and change Section "Conclusions" to "Conclusion and outlook".*

⇒ We have rearranged the paragraph as your suggestion.

**5. Conclusions and Outlook**

"The Noah Land Surface Model (Noah LSM) generally underestimates snow amount during the peak winter and shows earlier snow melting in spring, whereas it overestimates snow albedo (SA) over Eurasia, mainly due to uncertain parameterzation processes (Saha et al., 2017). Our experiment with no optimization (CNTL) reveals underestimation of snow depth (SD) and fractional snow cover (FSC) and overestimation of SA compared to the in-situ or satellite observations. Therefore, we have developed a coupled system of micro-genetic algorithm (micro-GA) and Noah LSM to reduce the uncertainties in parameterized snow processes through optimization of parameter values. This parameter estimation is an effort to further improve the model performance by reducing uncertainty in pre-existing parameterization schemes by optimizing the parameter values inside the schemes based on the observational data that reflect local characteristics to improve snow simulation. If the employed parameterization scheme has less uncertainty, improvement by parameter estimation on that scheme may not be significant; if the scheme has large uncertainty in parameter values, parameter estimation may bring about prominent improvement in the scheme's performance.

The coupling system of micro-GA and Noah LSM automatically estimates the optimal snow-related parameters by objectively comparing observations and model solutions through the fitness function. Instead of trial-and-error procedures, it has an advantage to reduce a substantial amount of computational time. The original micro-GA reduces the computational time using the elitism and re-initialization methods in the small number of individuals. However, we have developed a parallel system on the coupled system to further improve the computational efficiency in this study; it enables us to simultaneously execute multiple individuals in one generation and multiple Noah LSM runs in one individual.

Six parameters included in the snow processes in Noah LSM have been optimized by using a micro-GA during the period 2009-2018 in South Korea (SK). The first parameter is the distribution shape parameter that participates in the FSC calculation and shows a positive correlation with the FSC: the optimized value is expected to increase the FSC, but it is

not sufficient to alleviate its underestimation problems. The second parameter is snow water equivalent threshold value that implies 100 % snow cover and also is used in the FSC calculation depends on the land cover type: its optimized value improves the FSC in terms of RMSE and mean bias over some stations. The third parameter is the maximum SA coefficient: its optimized (decreased) value improves the RMSE by reducing the overestimation of SA. The fourth parameter is the coefficient in the maximum albedo of fresh snow, and its optimized value was similar to the default one. The other two parameters are related to the fresh snow density used for the SD calculation. In particular, the sixth parameter depends on air temperature and its optimization brings about the largest improvement in SD: the optimized (reduced) value remarkably reduces the RMSE, which ameliorates the underestimation problem of SD. This significant improvement of SD is due to the high spatial and temporal resolutions of observations.

The best combinations of snow parameters optimized for SK can be used to improve the snowfall prediction. Our results showed improvement in all snow variables in terms of RMSE by 3.3 %, 6.2 %, and 17.0 % for FSC, SA, and SD, respectively. Furthermore, SD increased after optimization, which lead to increases in both soil temperature and sensible heat flux via insulating response; soil moisture also increased due to increased SD in previous years. This implies that the optimized snow parameters not only let the model solutions close to the observations but also act in a physically consistent manner. Satellite observations proved to be effective in the optimization; however, their coarse resolution as well as insufficient number of stations used for optimization often restrict improvement of the snow variables, as shown in some discouraging statistics including the mean bias and the coefficient of determination ($R^2$).

Based on the encouraging optimization results in the off-line Noah LSM, we plan to optimize the Noah LSM in a coupled land-atmosphere prediction system. The online Noah LSM can produce a spatial distribution of model variables over the land surface, which allows a two-dimensional assessment of model performance and a three-dimensional extension through various interactions between the land surface and the atmosphere. We anticipate the optimized snow parameters can lead to positive effects on the atmospheric variables through the changes of heat fluxes as well as snow variables in Noah LSM. As a result, we can identify how optimal parameters are appreciated in SK in terms of both horizontal and vertical distributions. Furthermore, the micro-GA-Noah LSM coupled system can be utilized to optimize other parameters in Noah-LSM, including the ones that indirectly affect the snow processes."

**References**

[1] Rosolem, R., Gupta, H. V., Shuttleworth, W. J., de Gonçalves, L. G. G., and Zeng, X: Towards a Comprehensive Approach to Parameter Estimation in Land Surface Parameterization Schemes, Hydrol. Process., 27(14), 2075-2097, 2013.

---

## Author Response (AR4)

Response to Remarks from the Preceding Review File Validation for the
Manuscript gmd-2021-333
**"Optimization of Snow-Related Parameters in Noah Land Surface
Model (v3.4.1) Using Micro-Genetic Algorithm (v1.7a)"**
by Sujeong Lim, Hyeon-Ju Gim, Ebony Lee, Seungyeon Lee, Won Young Lee,
Yong Hee Lee, Claudio Cassardo, and Seon Ki Park

We appreciate your validation of our manuscript. We changed the colour
scheme to allow readers with colour vision deficiencies to correctly interpret our
findings in the manuscript (e.g., Figure 4(b), Figure 5, Figure 6 and Figure 7).
Accordingly, the plot scripts also changed in the Zenodo code.

We also conducted minor corrections and they are summarized in below (red
color represents the removed sentences while blue color represents the added
sentences):

1. L54: Results,  and conclusions and outlook are
   provided in sections 4 and 5,  respectively.

2. L59: soil moisture and soil temperature

3. Figure 1(c) and its caption: We corrected the x-axis of Figure 1(c) and its
   caption as $\alpha_{max,CofE}$.

4. L119-120: SA shows similar sensitivities to both parameters within the
   same range but is a bit more sensitive  to $C$.

5. L129: where $P_1 = 0.05$ g cm$^{-3}$ and $P_2 = 0.0017$ g cm$^{-3}$ $^{\circ}$C$^{-1}$ are the
   default values of the coefficients.

6. L137:  → individuals

7. L208:  → thicknesses

8. L249-250: We moved the L245-246 in the original manuscript to L249-250
   in the revised manuscript.

   → The uniform crossover in which each gene is selected randomly from
   one of the parent chromosomes makes all populations perform a crossover
   at every generation to acquire the diversity (Lee et al., 2005).

9. Figure 4(b): We labeled the land cover types (LCT) in the each colored
   line.

10. Figure 5: We changed the colour scheme.

11. L305-306: We removed the '(Table 4)' at end of sentence.

    → In the VRF_5, new parameter values — $P_s$, $\alpha_{max,CofE}$, $C$, $P_1$, and
    $P_2$ — optimized by the micro-GA result in an improvement of RMSE for
    FSC, SA and SD, such as 0.7 %, 5.4 % and 13.7 %, respectively

12. L312-313: We removed the '(Table 4)' at end of sentence.

    $\rightarrow$ Next, SD shows the greatest RMSE improvement of 13.7 % .

13. L325-330: To erase the statistics in OPT_W for each LCT which can induce confusion, we revised the sentences like below:

    ~~To supplement insufficient improvement in the FSC, we have additionally optimized the $W_{max}$ in function of LCT (OPT_W) based on the five parameters optimization results from OPT_5. Here, we have only used the FSC to define the fitness function, they not considering SA and SD. Therefore, the fitness function is defined using Eq. (8) where the $\vec{x}$ is only the FSC, so the normalized process with Eq. (9) is not needed. As a result, the OPT_W further improves the RMSE of FSC compared to previous optimization results in the DBF, MF, WS, and UB by 4.6 %, 11.9 %, 7.7 %, and 5.5 %, respectively, while weakly decreases by 0.1 % in CL. To solve the under-estimated FSC that occurred at all stations in VRF_5, we anticipate OPT_W decreases the $W_{max}$, which leads to an increase of FSC. Consequently, the OPT_W generates a decreased $W_{max}$ in the MF and UB and other LCTs (e.g., DBF, WS and CL) generate increased $W_{max}$.~~

    $\rightarrow$ To supplement insufficient improvement in the FSC, we have additionally optimized $W_{max}$ in function of LCT (OPT_W) using the optimized values of five parameters from OPT_5. Here, we have only used the FSC to define the fitness function, not considering SA and SD; thus, the fitness function is defined using Eq. (8) where the FSC is the only element of **x**, and the normalized process with Eq. (9) is not necessary. As a result, the OPT_W further improves the RMSE of FSC in VRF_6 compared to VRF_5 in most stations: the significant decreases in $W_{max}$ over MF and UB leads to an increase in the FSC, possibly alleviating the underestimation problem of FSC in VRF_5.

14. L332-333: When the optimized five parameters are used except the $W_{max}$ (VRF_5), SA and SD are improved, and FSC shows a weakly improvement in RMSE performance (Table 4).

15. L334-338: As a result, an improvement of RMSE for the FSC, SA, and SD is 3.3, 6.2, and 17.0 %, respectively . However, the MB for the FSC strengthens from 9.1 % to 11.9 % in VRF_6 (Table 4 and Fig. 5(a)) due to larger negative bias especially in the DBF. On the other hand, SA and SD reduce the MB against the CNTL and enhance the improvement ratio from 26.9 % to 31.0 % and from 35.9 % to 44.2 %, respectively (Table 4 and Fig. 5(b)-(c)).

16. The title of first column in Table 4:  $\rightarrow$ Experiments

17. L358-359:

→ Figure 7 shows temporal changes in the snow variables after parameter optimization by comparing their time series of the observations and the model simulations (CNTL versus VRF6) for DBF represented by UL.

18. L361: The bias patterns in Fig. 7 are consistent with those in Fig.6(a)-(c).

19. Figure 6: We changed the colour scheme and cation description.

→ Scatter plots of observations (OBS) and model results (LSM) for snow variables FSC (left panels), SA (middle panels), and SD (in cm; right panels) from the verification experiments — CNTL (red black dots), VRF_5 (blue dots), and VRF_6 (green orange dots), which are evaluated over different LCTs; (a–c) DBF represented by the station UL, (d–f) MF by GM, (g–i) WS by NG, (j–l) CL by BR, and (m–o) UB by SL.

20. Figure 7: We changed the colour scheme and cation description.

→ Time series of the snow variables for DBF (e.g., UL) from May 2009 to April 2018: (a) FSC, (b) SA, and (c) SD (in cm). Observations are in black gray dots and model results are in red black dots for CNTL and in green orange dots for VRF_6.

21. L427-429 in Author contributions: Because Sujeong Lim and Seungyeon Lee use the same initial (e.g., SL), we distinguish the initials as follow: Sujeong is SuL. and Seungyeon Lee is SeL.

22. L555-556: We removed the blank in the titles.

→ Saha, S. K., Sujith, K., Pokhrel, S., Chaudhari, H. S., and Hazra, A.: Effects of multilayer snow scheme on the simulation of snow: Offline Noah and coupled with NCEP CFS v2, J. Adv. Model. Earth Sy., 9, 271–290, 2017.